# Comparison of Large Eddy Simulations against measurements from the Lillgrund offshore wind farm

Ishaan Sood[1], Elliot Simon[2], Athanasios Vitsas[1], Bart Blockmans[1], Gunner Chr. Larsen[2], and Johan Meyers[1]

[1]Mechanical Engineering, KU Leuven, Celestijnenlaan 300, Leuven 3001, Belgium
[2]DTU Wind and Energy Systems, Technical University of Denmark, Frederiksborgvej 399, 4000 Roskilde, Denmark

**Correspondence:** Ishaan Sood (ishaan.sood@kuleuven.be)

**Abstract.** Numerical simulation tools such as Large Eddy Simulations (LES) have been extensively used in recent years to simulate and analyze turbine-wake interactions within large wind farms. However, to ensure the reliability of the performance and accuracy of such numerical solvers, validation against field measurements is essential. To this end, a measurement campaign is carried out at the Lillgrund offshore wind farm to gather data for the validation of an in-house LES solver. Flow field data is collected from the farm using three long-range WindScanners, along with turbine performance and load measurements from individual turbines. Turbulent inflow conditions are reconstructed from an existing precursor database using a scaling-and-shifting approach in an optimization framework, proposed so that the generated inflow statistics match the measurements. Thus, 5 different simulation cases are setup, corresponding to 5 different inflow conditions at the Lillgrund wind farm. Operation of the 48 Siemens 2.3 MW turbines from the Lillgrund wind farm is parameterized in the flow domain using an Aeroelastic Actuator Sector Model (AASM). Time-series turbine performance metrics from the simulated cases are compared against field measurements to evaluate the accuracy of the optimization framework, turbine model and flow solver. In general, results from the numerical solver exhibited a good comparison in terms of the trends in power production, turbine loading and wake recovery. For four out of the five simulated cases, the total wind farm power error was found to be below 5%. However, when comparing individual turbine power production, statistical significant errors were observed for 16% to 84% of the turbines across the simulated cases, with larger errors being associated with wind directions resulting in configurations with aligned turbines. While the compared flapwise loads in general show a reasonable agreement, errors greater than 100% were also present in some cases. Larger errors in the wake recovery in the far wake region behind the LiDAR installed turbines were also observed. An analysis of the observed errors reveals the need for an improved controller implementation, improvement in representing meso-scale effects and possibly a finer simulation grid for capturing the smaller scales of wake turbulence.

## 1 Introduction

Recent years have seen the emergence of wind-farm simulation tools that cover the whole chain from flow-coupled aeroelastic models to power-grid models. The complexity of these models ranges from analytical tools, which simplify wake expansion and merging, to complex Computational Fluid Dynamics (CFD) solvers which represent the turbines and their influence on

the surrounding flow field. Amongst all these numerical tools, Large Eddy Simulations (LES) have been extensively used in recent years to resolve detailed representation of the turbulent flow in and around large wind farms (Calaf et al., 2010; Lu and Porté-Agel, 2011; Archer et al., 2013; Martínez-Tossas et al., 2015; Ghaisas and Archer, 2016; Munters and Meyers, 2018; Lin and Porté-Agel, 2019). This increased detail in simulating the physics governing wind-farm flows has facilitated the study of wind-farm aerodynamics and enabled the analysis of phenomena like turbine-wake interactions, gusts, atmospheric stratification and the effect of wind farms on local wind climate. A comprehensive review of LES for wind farm simulations addressing the aforementioned effects can be found in the references (Mehta et al., 2014; Porté-agel et al., 2020). Additionally, LES has also been used to investigate and develop coordinated wind-farm control strategies, which could provide the benefits of power maximization, asset life extension and grid frequency regulation, thus improving the performance and capabilities of wind farms. (Goit and Meyers, 2015; Yılmaz and Meyers, 2018; Bossanyi, 2018; Boersma et al., 2019; Frederik et al., 2020). However, to give credibility to these studies it is essential to validate numerical solvers against reliable measurement data. While wind-tunnel experiments provide a useful avenue for testing, their accuracy in representing full-scale wind farms is limited due to the size and measurement constraints of wind tunnels (Bastankhah and Porté-Agel, 2017). Therefore, proper validation of wind-farm numerical models requires accurate reference data in the form of detailed flow field and performance measurements from existing wind farms. To this end, a measurement campaign was carried out at the Lillgrund wind farm, located 10 km off the coast of southern Sweden, as part of the Horizon 2020 TotalControl project. The measurement campaign made use of 3 long-range LiDARs, which measure the inflow conditions for the farm while also resolving the flow field in a part of the Lillgrund wind farm for wake measurements. The flow field data was supplemented by simultaneous power and structural-load measurements from individual wind turbines. The combination of LiDAR inflow field data, turbine performance data, wake data and loading data provide a unique data set for the validation of coupled flow and aeroelastic solvers.

In this work, SP-Wind, an in-house aeroelastic LES solver, which has previously been used extensively for wind-farm modelling and control optimization, is used to simulate the operation of the Lillgrund wind farm during the measurement campaign. While previous wind-farm validation studies have been carried out using LES (Wu and Porté-Agel, 2011, 2013; Nilsson et al., 2014; Wu and Porté-Agel, 2015; Draper et al., 2016; Simisiroglou et al., 2018), this work differs on three fronts. First, the atmospheric conditions at the Lillgrund site are recreated in the numerical domain by analyzing inflow LiDAR measurements. Second, instead of initializing the flow field from scratch, as is the convention for LES precursor simulations (Stevens et al., 2014), the current work proposes a framework for reusing a previously generated precursor flow database for matching the conditions during the measurement campaign, substantially reducing the associated computational costs and time for LES wind-farm validation studies. Third, the current study utilizes a novel Aeroelastic Actuator Sector Model (AASM) for parameterizing the turbine forces in the numerical domain (Vitsas and Meyers, 2016) . Compared to other actuator models such as the Actuator Disc Model (ADM) and the Actuator Line Model (ALM), the AASM has the advantage of accurately representing rotating turbine blades while allowing for coarser time steps through spatial and temporal filtering, and decoupling of the LES time step and the time step of the flexible multibody model that is a part of the AASM.

The present article is organized as follows: Section 2 details the specifics of the measurement campaign and available data sets. Section 3 presents the specifications of the numerical solver used in this study, while Section 4 outlines the optimization

framework developed to recreate the atmospheric conditions at Lillgrund using a previously generated precursor data set. A comparison of individual turbine performance results from the numerical solver against field measurements, and a wake deficit analysis are presented in Section 5. Finally, Section 6 outlines a summary of the validation and the challenges associated with LES validation studies of wind farms.

## 2 Lillgrund offshore measurement campaign

### 2.1 Lillgrund offshore wind farm

The Lillgrund offshore wind farm is located approximately $10\,\mathrm{km}$ off the coast of southern Sweden, just south of the Öresund Bridge, where average wind speeds are overall close to $8.5\,\mathrm{m\,s^{-1}}$ (Sebastiani et al., 2021). The wind farm contains 48 wind turbines (Siemens SWT-2.3-93) with a total capacity of 110 megawatts ($\mathrm{MW}$). The farm's turbines have a rotor diameter of 93 metres, hub height of 65 metres, and a tip height of 115 metres. The farm is known to suffer from performance losses, as the turbines originally intended for the farm were replaced by larger models, leading to a tighter layout when normalized by turbine diameter (Simisiroglou et al., 2018).

### 2.2 LiDAR measurements

During the measurement campaign from September 2019 to February 2020, three long-range WindScanners, i.e. pulsed scanning Doppler wind LiDARs (Vasiljević et al., 2016), were installed on the Lillgrund wind turbine transition pieces and used to measure the flow field both upstream and within the farm, in the layout shown in Figure 1.

#### 2.2.1 Inflow LiDAR

The inflow measuring system on turbine B08, "Vara", performed repeating Plan Position Indicator (PPI) sector-scans with a constant elevation angle of $8°$ and azimuth sweep of $60°$. The center-line of the arc scan was intended to lie parallel with the B-row of turbines, however no hard targets were visible from its install location to allow for a precise alignment. Instead, the system was coarsely aligned during installation, and subsequently, the static misalignment was determined and later corrected for using the turbine's calibrated nacelle direction signal. The wake-scan positions (red and blue areas) were determined through an optimization where the LiDAR positions and points along the wake transect were first defined, and a trajectory for each LiDAR was generated which matched the position and timing of both LiDARs, along with kinematic controls to control the scan speed, motor acceleration, etc. The scanned areas are mostly symmetric, but not exactly due to the actual installed positions and wake transect point locations chosen.

The inflow LiDAR data was processed firstly by removing periods of low signal quality, i.e periods with Carrier to Noise Ratio (CNR) below $-26\,\mathrm{dB}$. Partial scans with a low proportion (<80%) of valid radial speed values were also filtered out. The remaining valid data was reconstructed into two-component horizontal wind vectors using the integrating Velocity-Azimuth Process (iVAP) method (Liang, 2007). Lastly, the direction misalignment due to imperfect installation of the system was cor-

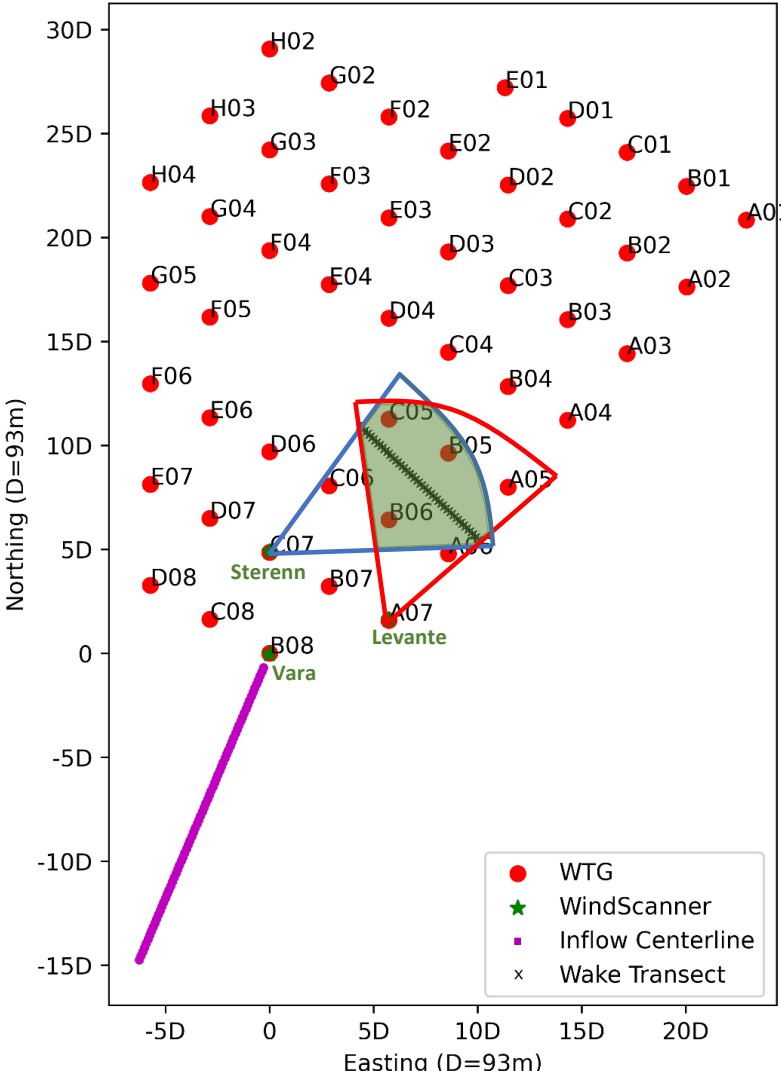

**Figure 1.** Top-down view of the the Lillgrund site and scan areas for inflow and wake measuring LiDARs installed on turbines B08 "Vara", A07 "Levante" and C07 "Sterenn". Coordinates are relative to the turbine B08 position, spaced in multiples of 5 rotor diameters. Vara performs PPI sector scans which are reconstructed into wind vectors along the magenta inflow centerline. Sterenn and Levante perform coordinated dual-Doppler complex trajectory scans which provide time and space synchronized measurements at 3 heights along the wake transect indicated with black crosses. The overlapping area between the areas scanned in red by Levante and blue by Sterenn are shaded in green. The positions within the green area which lie off the synchronized wake transect line are resolved through time-averaging and applying dual-Doppler retrieval to the 10-minute averaged wind field.

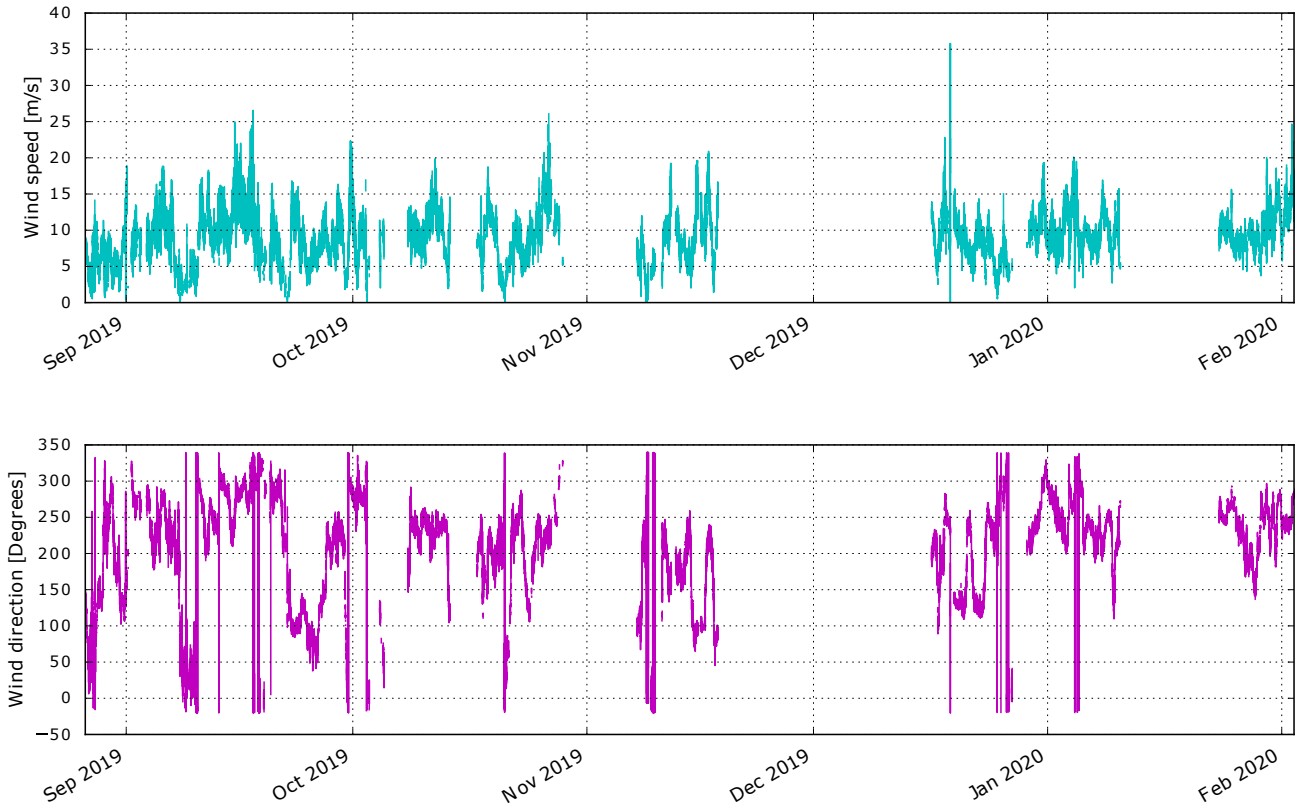

**Figure 2.** Wind-speed (top) and direction (bottom) measurements at hub height of 65 m processed from the inflow LiDAR mounted on turbine B08. Missing data corresponds to equipment downtime and filtering of low signal quality periods.

rected by determining the static offset between the nacelle direction measurements from turbine B08 and LiDAR reconstructed

wind direction. The static direction misalignment was found to be $-20.47°$. This method does not account for the fact that the turbine yaw controller has a delay in wind direction tracking due to actuator limits. Taken on average over the entire campaign however, it provides the best estimation possible when a hard target alignment is not feasible to perform. Note that this correction has no effect on the wind speed result from the LiDAR measurements, only the resolved wind direction.

A time-series of processed wind speed and direction inflow measurements corresponding to the turbines' hub height (65 m)

is shown in Figure 2, with the equivalent wind rose shown in Figure 3. This location corresponds to the LiDAR range gate at a distance of 430 meters, given the LiDAR's inclined beam and origin at 5 meters above mean-sea-level. Range gates between 70 and 1490 meters were sampled in steps of 20 meters.

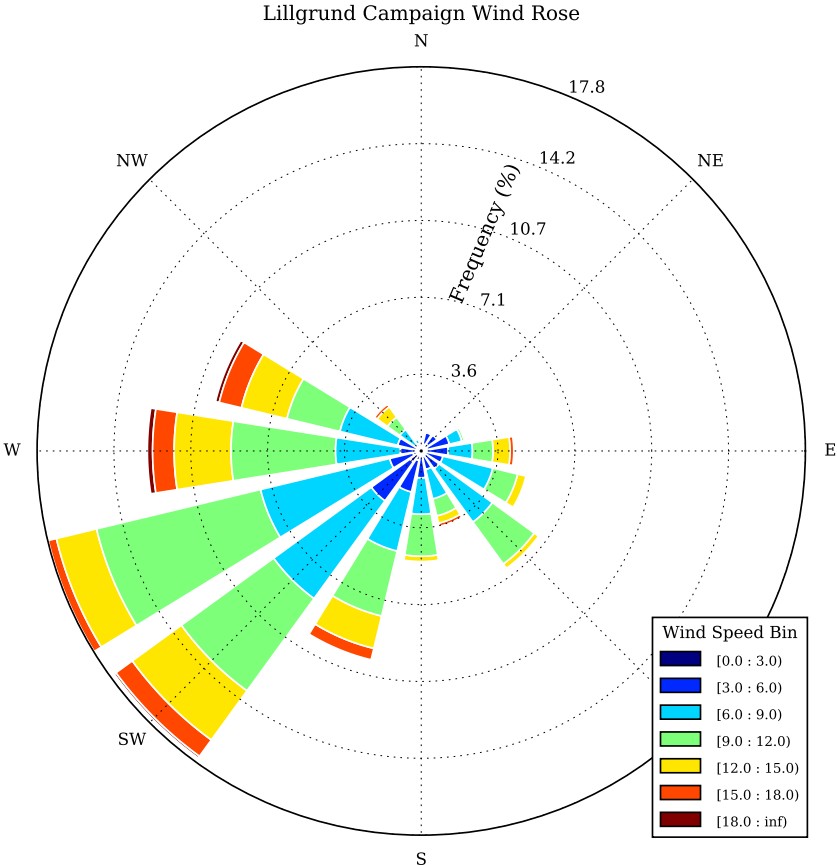

**Figure 3.** Wind rose depicting hub height wind speed and direction from the inflow measuring LiDAR.

### 2.2.2 Wake Scanning LiDARs

Two additional scanning LiDARs identical to the inflow measuring system were installed within the wind farm to measure
wake effects and intra-farm flows. The two wake scanning LiDARs designated as ("Levante" on A07 and "Sterenn" on C07)
performed coordinated dual-Doppler complex trajectory scans within the green overlapping area shown in Figure 1. The points
placed along the wake transect path indicated with black crosses were repeated at three heights (6,70,130 meters ASL) and
were time and space synchronized using the DTU WindScanner software. Overlapping scan areas which lie away from the
wake transect path are not time-space synchronized but have been averaged over a 10-minute period to produce a 3D wind
field. From this, a horizontal slice has been taken to obtain the 2D wind field at constant hub height. The dual-Doppler wind
retrieval method used in this study follows the process outlined in Simon and Courtney (2016).

## 2.3 SCADA data

Data from the wind farm monitoring system was provided by Vattenfall. This included the following channels from all 48 turbines in the wind farm: Active Power, Blade Pitch Angles (A/B/C), Nacelle Direction, Rotor RPM, and Wind Speed. The raw streaming data format did not maintain a specific time resolution, where deviations in the sampling rate existed between channels and over time. The data set was processed into a constant sampling rate of $0.5\,\text{Hz}$ timesteps across all channels. This period was chosen as it was the highest time-resolution present in the raw data files. The periods and channels with lower time-resolutions were upsampled (interpolated) to match. The raw nacelle direction signals were found to have differing offsets, and were then calibrated on a per-turbine basis and corrected in the final data set. Given the fact that met mast data was not available, the calibration was performed using sets of turbines each consisting of a front turbine and the corresponding closest downstream turbine. The inflow direction generating the largest wake loss at the downstream turbine location corresponds to the direction from the front turbine to the downstream turbine, whereby the offset of the measured nacelle direction at the front turbine follows directly. To cover the complete wind rose, 4 direction sectors of 90 deg. was defined, and within each of these the nacelle direction calibration was performed using 2-3 sets of turbines.

## 2.4 Load data

Six wind turbines (designations: B06, B07, B08, C08, D07 and D08) have been outfitted in the past with load measuring equipment, and could be used during the TotalControl measurement campaign. These data are available starting November 2019 at 10Hz sampling frequency. Strain gauge measurements are available both at the blade root (installed 1.5 meters from blade root) and tower base (installed 8.52 meters from tower base). The tower base position includes two sensors oriented 90 degrees apart. The sensors were installed and calibrated by Siemens Gamesa.

## 3 Numerical solver

### 3.1 Fluid solver

SP-Wind is a wind-farm Large Eddy Simulation code built on a high-order flow solver developed over the last 15 years at KU Leuven (Calaf et al., 2010; Allaerts and Meyers, 2015; Munters and Meyers, 2018). The three-dimensional, unsteady, and spatially filtered Navier-Stokes momentum and temperature equations

$$\frac{\partial \tilde{\mathbf{u}}}{\partial t} + (\tilde{\mathbf{u}} \cdot \nabla)\tilde{\mathbf{u}} = -\frac{\nabla(\tilde{p} + p_\infty)}{\rho} - \nabla \cdot \boldsymbol{\tau_s} + 2\boldsymbol{\omega} \times \tilde{\mathbf{u}} + g\frac{(\tilde{\theta} - \theta_0)}{\theta_0} + \mathbf{F_{tot}} \tag{1}$$

$$\frac{\partial \tilde{\theta}}{\partial t} + (\tilde{\mathbf{u}} \cdot \nabla)\tilde{\theta} = -\nabla \cdot \boldsymbol{q_s} \tag{2}$$

are solved. In these equations, $\tilde{\mathbf{u}} = [\tilde{u_1}, \tilde{u_2}, \tilde{u_3}]$ is the filtered velocity field. Further, $\tilde{\theta}$ is the filtered potential temperature field, and $\theta_0$ is the background adiabatic base state. The pressure gradient is split into a mean background pressure gradient $\nabla p_\infty$

driving the mean flow, and a fluctuating component $\nabla \tilde{p}$. The very high Reynolds numbers in the atmospheric boundary-layer flow combined with typical spatial resolutions in LES justify the omission of resolved effects of viscous momentum transfer and diffusive heat transfer. Instead, these are represented by modeling the subgrid-scale stress tensor $\boldsymbol{\tau_s}$ and the subgrid-scale heat flux $q_s$ originating from spatially filtering the original governing equations (Allaerts and Meyers, 2015). Coriolis effects are included through the angular velocity vector $\boldsymbol{\omega} = \Omega sin\phi$, where $\Omega$ is the earth's rotation and $\phi$ is the latitude of the wind farm. Thermal buoyancy is represented by $g(\tilde{\theta} - \theta_0)/\theta_0$, with $g$ the gravitational acceleration, $\tilde{\theta}$ the filtered potential temperature and $\theta_0$ a reference temperature. The effect of surface friction is included using a standard wall-stress model, corresponding to a logarithmic velocity profile with a roughness length $z_0$ (Bou-Zeid et al., 2005) to represent marine boundary layers, without the need to resolve wave effects on the mesh. Finally, $\mathbf{F_{tot}}$ represents body forces added to the flow and consists of $\bar{\mathbf{F}}$ and $\mathbf{F_R}$, which represent forces exerted by the wind turbines on the flow and a fringe-region approach to introduce inflow boundary conditions in the domain respectively.

Spatial discretization is performed in the horizontal and span-wise directions by using pseudo-spectral schemes, while vertical fourth-order energy-conservative finite differences are used in the vertical direction. The equations are marched in time using an explicit fourth-order Runge-Kutta scheme, and grid partitioning is achieved through a scalable pencil decomposition approach. Subgrid-scale stresses are modeled with a standard Smagorinsky model with Mason and Thomson wall damping (Allaerts and Meyers, 2015).

## 3.2 Structural solver

Deformation of the turbine blade and tower is employed by a finite-element floating frame of reference formulation (Shabana, 2013). Each element is described by reference coordinates which specify its position and orientation, and elastic coordinates that define its deformation with respect to the body coordinate system. Bryant angles are used to describe the orientation of the rotor's body reference frame, however only the rotation of the turbine rotor is assumed to contribute to its dynamic behaviour. Deformations along the tilting, yawing and pre-coned axis are taken into account quasi-statically. The governing equation for the system can be written as (Shabana, 2013)

$$\mathbf{M}(\boldsymbol{q})\ddot{\boldsymbol{q}} + \mathbf{C}\dot{\boldsymbol{q}} + \mathbf{K}(\boldsymbol{q})\boldsymbol{q} + \boldsymbol{\Phi}_q^T \boldsymbol{\lambda} = \boldsymbol{Q_a} + \boldsymbol{Q_g} + \boldsymbol{Q_v} \tag{3}$$

$$\boldsymbol{\Phi}(\boldsymbol{q}) = 0 \tag{4}$$

where, $\mathbf{M}$, $\mathbf{C}$, $\mathbf{K}$, are the mass, damping and stiffness matrices respectively, computed using the structural specifications of the Siemens SWT-2.3-93 turbines. The vector $\boldsymbol{q}$ represents the generalized coordinates, while $\dot{\boldsymbol{q}}$ and $\ddot{\boldsymbol{q}}$ represent their first and second time derivatives. $\boldsymbol{\Phi_q}$ and $\boldsymbol{\lambda}$ are the constrain Jacobian matrix and Lagrange multipliers, respectively, and $\mathbf{Q_g}$ represents the gravitational loads acting on the rotor and tower elements. The vector $\boldsymbol{Q_a} = \begin{bmatrix} F_A^{rtr} & F_A^{twr} \end{bmatrix}$ contains the aerodynamic loads evaluated at the rotor and tower nodes, as described in section 3.3. Finally, $\mathbf{Q_v}$ is composed of the Coriolis and gyroscopic loads (Shabana, 2013). Further details regarding the coordinate system used and the derivation of the equations of motion is given in Appendix A.

To solve the equations of motion, first an eigenvalue problem is solved without damping and external loading to extract the mode shapes and natural frequencies of the structure. Then, the order of the system is reduced by a common modal transformation technique. Hence, the rotor blades are represented by 6 modes (two flap wise, two edgewise, one torsional and one axial) and the tower is represented by 4 modes( two side-to-side and two fore-aft). Finally, the reduced order system is integrated in time by using the generalized-$\alpha$ method, with a time-step of $0.01\,\mathrm{s}$ and spectral radius of 0.9 for low numerical damping (Arnold et al., 2007).

## 3.3 Turbine Model

As it's computationally prohibitive to fully resolve a wind turbine structure and its forces, actuator methods have been extensively used in research to parameterize wind turbine forces onto the flow grid. The most widely used amongst these models is the Actuator Disc Model (ADM) and the Actuator Line Model (ALM) (Churchfield et al., 2017). The ADM is a simple representation of a wind turbine in the flow domain, in which a disc is used to parameterize the turbine forces on the flow. Though simple, the ADM does not accurately represent wind turbine operation due to a lack of discrete rotating turbine blades, and hence often requires additional tuning to improve performance. Comparatively, the ALM provides a more accurate parametrization by representing the turbine blades as actuator lines, with each point exerting a force on the flow based on its local inflow velocity and airfoil distribution. However, the ALM suffers from a limitation that the movement of the actuator line tip over a time step is limited to the size of a cell, thus requiring very fine time steps and hence high computational costs. To overcome this limitation, an Actuator Sector Model (ASM) was developed, which swept the rotor forces across a sector area and hence allowed for coarser time steps. The ASM therefore represents an intermediary between the ALM and ADM, with the ability to resolve structures in the near wake with greater detail than the ADM due to the presence of rotating actuator lines, while avoiding the high computational cost associated with the fine time steps of the ALM (Storey et al., 2015). The ASM was later extended to an Aeroelastic Actuator Sector Model (AASM) by incorporating two-way Fluid Structure Interaction (FSI) coupling, to account for structural deformations (Vitsas and Meyers, 2016).

In this work, the Siemens $2.3\,\mathrm{MW}$ turbines are modeled by using the AASM, coupled with the nonlinear flexible multibody dynamics model described in the previous section. Since the LES computations are more intensive than integrating the structural equations, a sub-cycling process is employed, for which the aeroelastic coupling scheme is shown in Figure 4. The relative velocity $V_{rel}$ is evaluated at each airfoil element along the blade based on the induced velocity field $\hat{u}_x$ at the airfoil's deflected position, and on the blade's out-of-plane and in-plane motion represented by $\dot{q}_{O_oP}$ and $q_{iP}$ respectively. The relative velocity $V_{rel}$ also includes the effect of rotor angles (yaw, tilt and precone) by using rotation matrices to transform the incoming flow field. The flow angle $\phi$ is then determined from the involved velocity triangle, which comprises of the the pitch angle $\beta$, the torsional deflection $\tau$ and the angle of attack $\alpha$. The lift $L$, drag $D$ and pitching moment $M$ at each airfoil section is then determined using 2D airfoil look-up tables, and used to evaluate the aerodynamic forces $\mathbf{F_A^m}$ comprising of the normal, tangential and span-wise forces, $F_N^m$, $F_T^m$ and $F_S^m$. This is done at every sub-cycle $m$, and serves as an input to the equations of motion given by equation 3, which are subsequently solved at each sub-step. The forces are then spatially and temporally filtered to

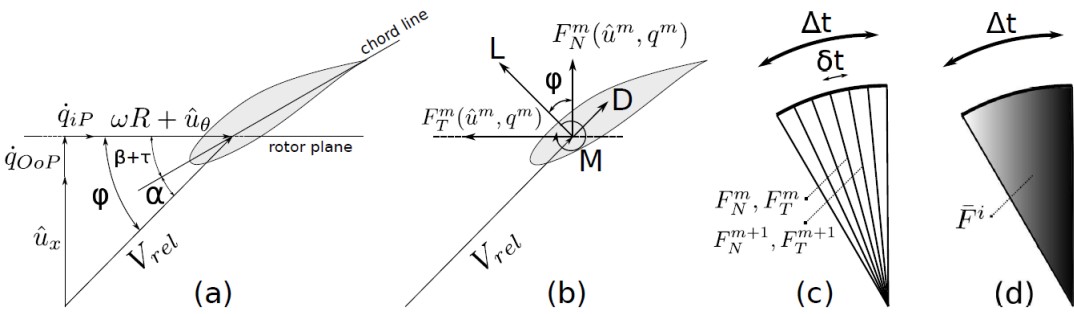

**Figure 4.** The AASM comprises of the steps: (a) evaluation of the angle of attack from the airfoil's cross-section velocity triangle and blade's motion, (b) the local cross-section forces are computed from 2D airfoil data, (c) the blades sweep a sector area using a sub-cycling scheme, and (d) the sector area forces are time-filtered. (d) denotes that more weight is given to the forces in the end position. Figure taken from Vitsas and Meyers (2016).

obtain the body forces $\mathbf{F}$, which serve as an input to the flow solver in equations 1. Further details of the coupling of the turbine model, flow solver and filtering are given in Appendix B.

The pitch and rotational speed of all the turbines are controlled using an implementation of the DTU wind energy controller (Hansen and Henriksen, 2013). A comparison of the simulated power output and thrust in SP-Wind using AASM under the influence of a range of uniform velocities against reference data for the Siemens $2.3\,\mathrm{MW}$ turbines is shown in Figure 5. Slight differences can be seen in both the simulated power and thrust, which can be attributed to the coarse grid resolution across the turbine blades in SP-Wind, which was was restricted due to the resolution of the precursor simulations used as they were originally developed for a bigger 10MW turbine and a larger wind farm. Using the same grid resolution for the smaller 2.3 MW turbines at Lillgrund leads to differences in induction at the rotor plane.

## 4 Recreating the inflow conditions at Lillgrund

### 4.1 Precursor database

The turbulent inflow conditions for wind-farm inflow are obtained from the publicly available precursor data from the TotalControl Flow database (Munters et al., 2019a, b, c, d). The precursor data contains unsteady three-dimensional flow data of an unperturbed atmospheric boundary layer (i.e. without the influence of turbines). The database comprises of two Pressure Driven Boundary Layers (PDBL) and three Conventionally Neutral Boundary Layers (CNBL), spanning different surface roughness lengths and boundary layer heights. The TotalControl flow fields are initialized using mean velocity profiles upon which random divergence-free perturbations are added. These initial conditions are then advanced in time for 20 physical hours, so that the influence of the unphysical perturbations has disappeared, and the flow has reached a fully turbulent and statistically stationary state. The PDBL is a simple representation of a neutral atmospheric boundary layer which ignores the

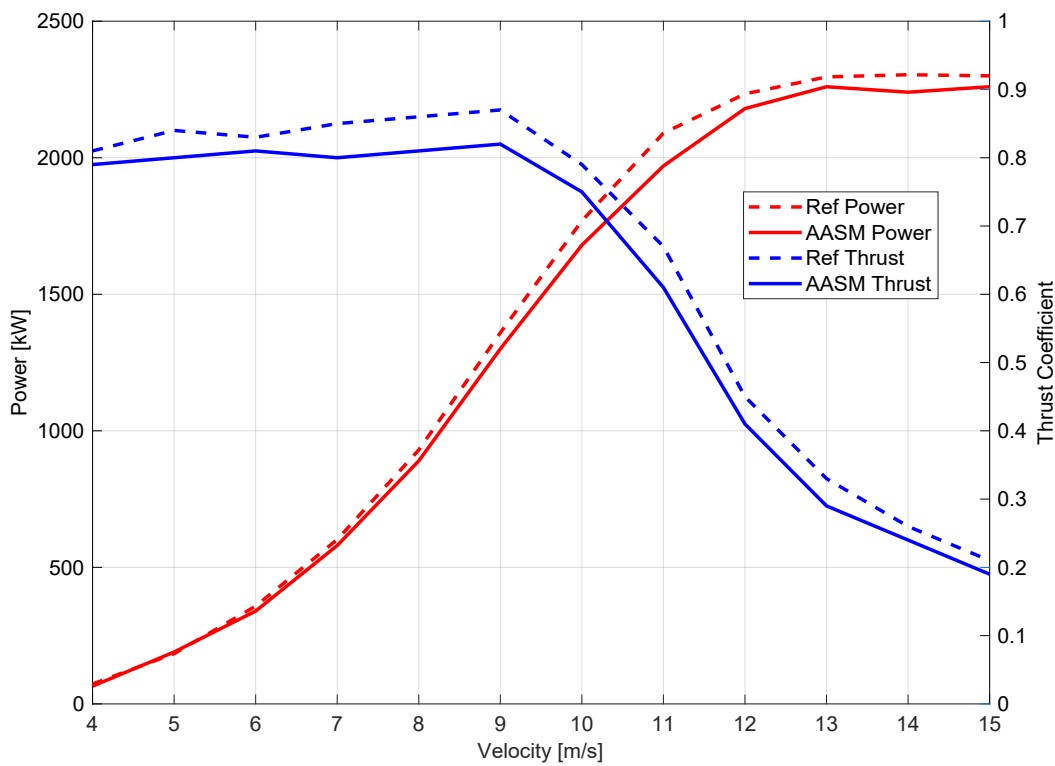

**Figure 5.** Comparison of simulated power and thrust coefficient in SP-Wind using the AASM against reference data (Göçmen and Giebel, 2016)

.

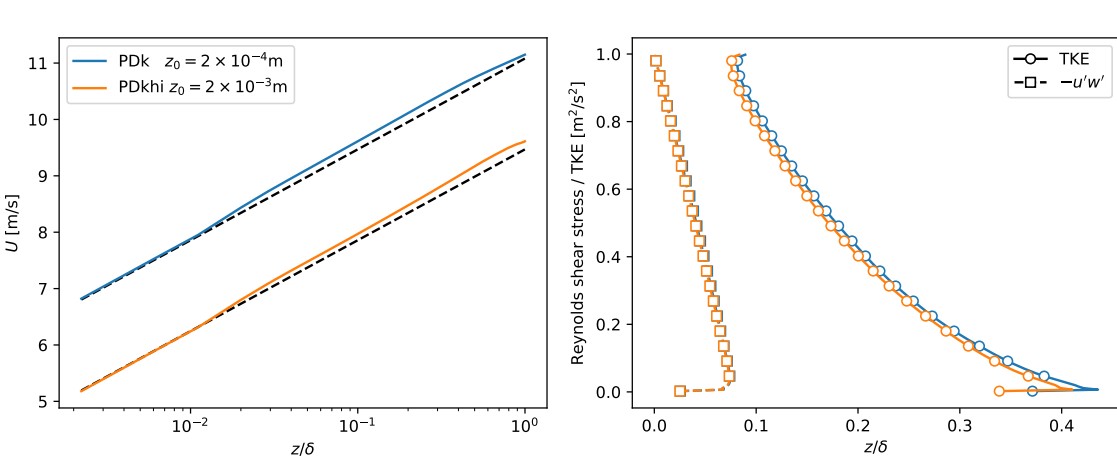

**Figure 6.** Flow profiles for PDBL cases. Left: Mean velocity. Dashed lines indicate log-law profiles. Right: resolved Reynolds shear stress and turbulent kinetic energy. Markers are plotted every 10 data points for clarity.

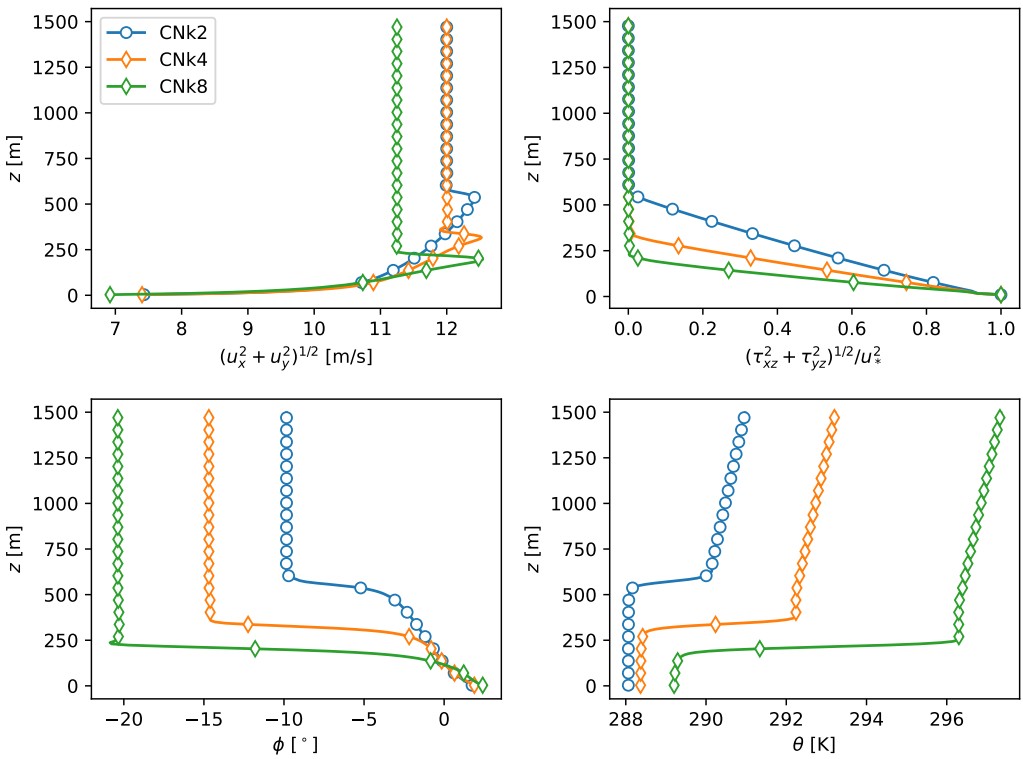

**Figure 7.** Flow profiles for CNBL cases. Top left: Horizontal Velocity. Top right: Total (Resolved + Subgrid) shear stress. Bottom left : Wind veer. Bottom right : Potential temperature. Markers are plotted every 10 data points for clarity. Figure taken from Sood et al. (2020a).

effects of rotation and thermal stratification, while CNBL provides a more realistic representation by including these effects. Both these boundary layer types have been extensively used in wind-farm LES. Specifications of the five different boundary layers are given in Table 1, while their flow profiles are shown in Figures 6 and 7. The database has previously been used in a study to determine the effect of CNBL height on wind-farm performance (Sood et al., 2020b). All the boundary layers

have been initialized using a friction velocity of $0.28\,\mathrm{m\,s^{-1}}$, which is a typical value for marine boundary layers (Brost et al., 1982). For the PDBL, this requires a driving pressure gradient of $\nabla p_\infty/\rho = -5.2267 \times 10^{-5}\,\mathrm{m\,s^{-2}}$ according to the equation $u_* = \left(-\frac{H}{\rho}\nabla p_\infty\right)^{1/2}$. Further information regarding the initialization of the boundary layers and the precursor database is available in the provided references (Anderson et al., 2020; Allaerts and Meyers, 2015). A stream-wise slab of the velocity and temperature field was stored to disk when running the precursor, and is later introduced in the wind-farm domain by means of

body forces in a so-called fringe region (Stevens et al., 2014; Munters et al., 2016). To match the inflow conditions measured by the LiDAR measurement campaign, the data from the precursor data set can be transformed to different flow conditions by re-scaling and shifting the flow variables. Using friction velocity $u_*$ for velocity scaling, different wind speeds can be attained by re-scaling the entire flow field by a different target friction velocity $u_*^t$. This is possible for offshore wind farms, as the

**Table 1.** Specifications of the TotalControl flow database (Anderson et al., 2020; Munters et al., 2019a, b, c, d)

| Case | Boundary layer height | Surface Roughness | Capping Inversion strength |
|------|-----------------------|-------------------|----------------------------|
| *PDk* | 1500 m | $2 \times 10^{-4}$ m | - |
| *PDkhi* | 1500 m | $2 \times 10^{-3}$ m | - |
| *CNk2* | 500 m | $2 \times 10^{-4}$ m | 2 K |
| *CNk4* | 250 m | $2 \times 10^{-4}$ m | 4 K |
| *CNk8* | 125 m | $2 \times 10^{-4}$ m | 8 K |

solution is scale invariant at high Reynolds numbers. Additionally, in line with the classical outer layer similarity hypothe-
sis (Townsend, 1976), for offshore atmospheric boundary layers at high Reynolds numbers the roughness elements are much
smaller than the boundary-layer height, and hence the roughness acts merely to increase surface stress without any structural
changes in the flow (Castro, 2007; Jiménez, 2004). The effect of a different target roughness lengths $z_0^t$ can thus be imposed
by applying an offset on the mean flow in line with the difference in surface roughness. Hence, denoting the imposed reference
friction velocity and roughness length in the current cases by $u_*^r$ and $z_0^r$ respectively, the flow can be re-scaled and shifted as

$$\mathbf{u^t}(\boldsymbol{x},t) = u_*^t \left[ \frac{\mathbf{u^r}(\boldsymbol{x},t)}{u_*^r} + \frac{1}{\kappa} \ln \frac{z_0^r}{z_0^t} \mathbf{e_1} \right] \tag{5}$$

Where, $\kappa$ is the Von Karman constant and $\mathbf{e_1}$ is a unit vector in the positive x direction, symbolizing that the flow shifting is
applied to the velocity $\tilde{u}_1$ in the stream-wise direction. The reference velocity field which is transformed, $\mathbf{u^r}(\boldsymbol{x},t)$, consists of
the stream-wise and span-wise velocities at time instant $t$. The same friction velocity and surface roughness is used for both
the velocity components. Scaling the velocity also leads to scaling of the time scales, according to the following equation

$$\Delta t^t = \frac{\Delta t}{u_*^t} u_*^r \tag{6}$$

For the CNBL data sets, a similarity parameter $h_* = |f_c| h / u_*$ relates the height of the atmospheric boundary layer with the
Rossby-Montogmery scale $u_* / |f_c|$ (Arya, 1975, 1978). Furthermore, an empirical relation for the height of the boundary layer
with the strength of the capping inversion is given as

$$h = A \frac{\theta_0}{g \Delta \theta} u_*^2, \tag{7}$$

where A is an empirical parameter with the value 500 and $\Delta \theta$ is the strength of the capping inversion (Csanady, 1974). Based
on the above relations, re-scaling via a new $u_*$ implies a change in latitude and $\theta_0 / \Delta \theta$, if $h$ is to remain constant.

## 4.2 Optimization framework

While SP-Wind does support changing wind directions during a simulation run (Munters et al., 2016), each simulation is
restricted to a single wind direction $\theta$ and a time frame of 75 minutes to limit computational costs. Thus, the available inflow

data from the measurement campaign was divided into numerous 75 minutes overlapping time windows. We define $\mathbf{P}$ as one of the LES data sets from the five available precursor cases listed in Table 1, and $\mathbf{Q}$ as a data set consisting of a 75 minute time block extracted from the available LiDAR database, with both $\mathbf{P}$ and $\mathbf{Q}$ containing only the stream-wise and span-wise velocity time series as the LiDAR measurements did not include vertical velocity measurements. Additionally, the velocities in the LES data sets are extracted from the thee dimensional TotalControl inflow database domain at the equivalent range gate locations as the LiDAR field measurements. Rows in the two data sets represent range gate locations from the measurement campaign, while columns represent time-series data. While various methods exist to compare the similarity between multi-dimensional time series data (Salarpour and Khotanlou, 2018), we limit our analysis to two simple distance metrics to determine the differences between the LiDAR measurements and the available LES precursors. These metrics are defined as,

$$d_1(\mathbf{P},\mathbf{Q}) = \|\overline{\mathbf{P}} - \overline{\mathbf{Q}}\|_{\boldsymbol{w}}^2. \tag{8}$$

$$d_2(\mathbf{P},\mathbf{Q}) = \|\text{cov}(\mathbf{P}) - \text{cov}(\mathbf{Q})\|_{\boldsymbol{w}}^2. \tag{9}$$

The first metric $d_1$ is a Euclidean norm which provides a measure of the differences between the time averaged profiles at range gate locations with the over bar denoting 75 minute time averages. The second metric $d_2$ , a Frobenius norm, is a measure of the differences between the co-variances of the two data sets, accounting for spatial correlations and also provides a measure for the differences in turbulence intensities across the rotor area. Without the inclusion of the covariance distance, it was observed that while the mean velocity profiles in obtained solutions matched well, they had large differences in the turbulence intensities. The metrics are also assigned weights according to a vector $\boldsymbol{w}$ as shown in Figure 8 (a), giving highest preference to the range gates spanning the rotor area 20% around the hub height, followed by the remaining rotor area and finally the rest of the vertical domain. As discussed in Section 4.1, the available LES data sets can be modified for each case $\mathbf{P}$ by using the scaling and shifting parameters $u_*^t/u_*^r$ and $z_0^t/z_0^r$, henceforth collectively referred to as the transformation vector $\boldsymbol{\zeta} = [u_*^t, z_0^t]$ , to obtain a new LES realization denoted by $\mathbf{P}_\zeta$ . Therefore, the distance metrics $d_1(\mathbf{P}_\zeta,\mathbf{Q})$ and $d_2(\mathbf{P}_\zeta,\mathbf{Q})$ can be determined over the entire 6 month measurement campaign between each LES data set $\mathbf{P}$ and a LiDAR data set $\mathbf{Q}$, for different combinations of the transformation parameters $\boldsymbol{\zeta}$. The sum of both the metrics can be used as a measure of similarity between the LES and LiDAR data, with lower values indicating greater similarity. Thus for each data set $\mathbf{P}$, a minimization problem can be defined to determine the transformation vector $\boldsymbol{\zeta}$ and the time window $\mathbf{Q}$ from the measurement campaign which returns the least distance between the LES data and LiDAR measurements, indicating highest similarity. The cost function of the optimization problem can be defined as,

$$\min_{\boldsymbol{\zeta},\mathbf{Q}} \; d_1(\mathbf{P},\mathbf{Q},\boldsymbol{\zeta}) + d_2(\mathbf{P},\mathbf{Q},\boldsymbol{\zeta}) \tag{10}$$

and is solved using the SLSQP solver from the SciPy Python package (Virtanen et al., 2020). As the distances are calculated for both the stream-wise and span-wise velocity components, the wind direction error is implicitly accounted for and minimized. In its current form, both the distance metrics are given equal weights in the minimization problem, however future work could

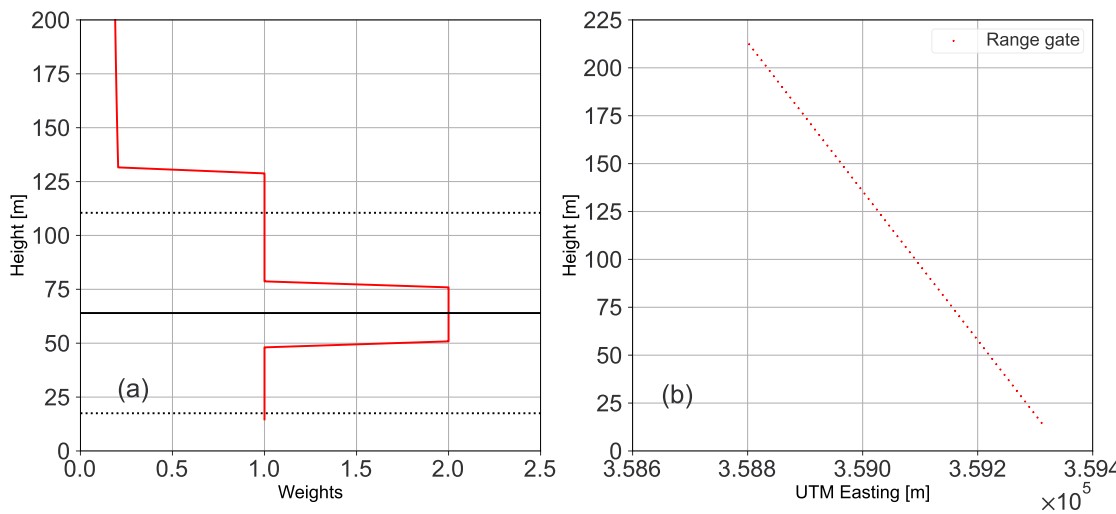

**Figure 8.** (a) Height varying weights used for the distance metrics in the minimization problem. (b) Locations of 72 range gates, with range gate 1 being the lowest and closest to the turbine B08, and 72 being the highest and furthest away.

**Table 2.** Specifications of selected validation cases

| Case | Measurement campaign time | Friction velocity ($u_*$) | Surface roughness ($z_0$) | hub height wind direction ($\theta$) | hub height velocity ($S$) |
|------|---------------------------|---------------------------|---------------------------|--------------------------------------|---------------------------|
| $PDk_1$ | 2019-12-23,T01:14:44 | $0.267\,\mathrm{m\,s^{-1}}$ | $2.17 \times 10^{-4}$ | $119°$ | $8.2\,\mathrm{m\,s^{-1}}$ |
| $PDk_2$ | 2019-12-18,T18:48:04 | $0.275\,\mathrm{m\,s^{-1}}$ | $2.00 \times 10^{-4}$ | $243°$ | $8.5\,\mathrm{m\,s^{-1}}$ |
| $PDk_3$ | 2019-09-24,T18:01:46 | $0.168\,\mathrm{m\,s^{-1}}$ | $5.60 \times 10^{-4}$ | $110°$ | $4.8\,\mathrm{m\,s^{-1}}$ |
| $CNk4_1$ | 2020-01-29,T05:40:36 | $0.280\,\mathrm{m\,s^{-1}}$ | $2.00 \times 10^{-4}$ | $251°$ | $10.8\,\mathrm{m\,s^{-1}}$ |
| $CNk8_1$ | 2020-01-07,T17:46:26 | $0.280\,\mathrm{m\,s^{-1}}$ | $2.00 \times 10^{-4}$ | $222°$ | $10.2\,\mathrm{m\,s^{-1}}$ |

explore a Pareto front for determining optimal weights for the two metrics. After sweeping through the entire measurement campaign, five unique time windows of 75 minute length each, corresponding to five different LES flow realizations which best matched the LiDAR data are obtained. The first three matches are obtained by transforming the *PDk* TotalControl LES data set, while the fourth and fifth matches are obtained from the *CNk4* and *CNk8* data set respectively, without any transformations. While additional matches were identified, which also contained flow realizations obtained from the *PDkhi* and *CNk2* LES data sets, we restrict further analysis to the best five cases with highest similarity due to computational limitations. A comparison of the mean vertical profiles at range gate location for these cases are shown in Figure 9 and their specifications are outlined in Table 2. A comparison of the average turbulence intensity across the range gates spanning the rotor area is also presented in Table 3. From Figure 9, it can be seen that majority of the selected LES cases match the LiDAR measurements within error

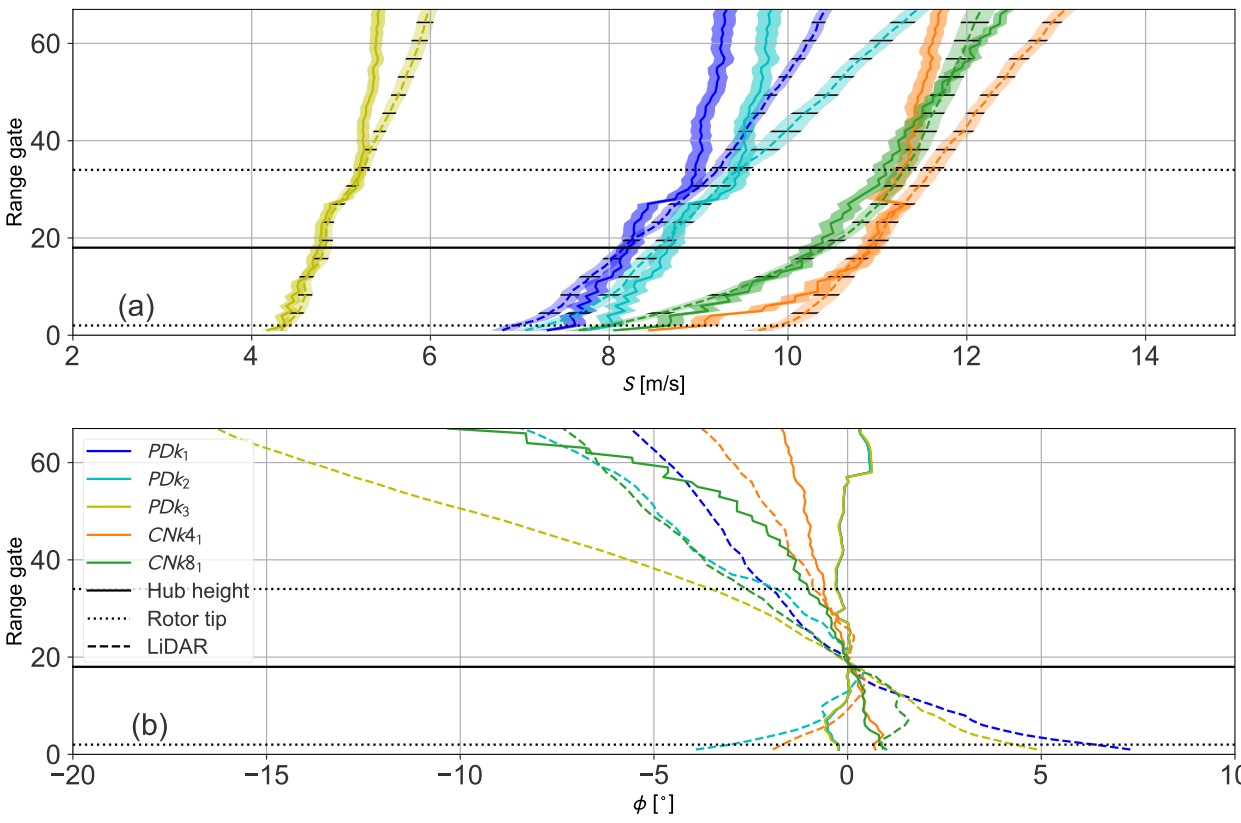

**Figure 9.** Comparison of vertical mean total velocity profile (a) and wind veer (b) between the selected LES and corresponding LiDAR data. Shaded area represents 95 percent confidence intervals on the mean, evaluated using the block bootstrap method.

**Table 3.** Turbulence Intensity (TI) comparison of validation cases, averaged over the rotor area. Error estimates represent 95% confidence interval and are calculated using the block bootstrap method with block length of 600 and 1000 bootstrap iterations.

| Case | LiDAR TI [%] | LES TI [%] |
|------|--------------|------------|
| $PDk_1$ | $5.52\pm0.09$ | $6.31\pm0.17$ |
| $PDk_2$ | $5.99\pm0.09$ | $6.27\pm0.14$ |
| $PDk_3$ | $6.64\pm0.11$ | $6.83\pm0.12$ |
| $CNk4_1$ | $5.12\pm0.17$ | $5.59\pm0.15$ |
| $CNk8_1$ | $5.86\pm0.16$ | $5.73\pm0.21$ |

limits across the rotor area. As the sample size of comparison is limited, bootstrapping is used to determine the measure of accuracy of the computed means. Since a traditional bootstrap approach of randomly re-sampling the original time-series data is inappropriate for time series with intrinsic correlation, the moving block bootstrap method is utilized(Kunsch, 1989). The length of individual blocks in the moving bootstrap method was set to 10 minutes, as a compromise between having enough

bootstrap blocks from the measured time-series data, and keeping the block lengths large enough to ensure that individual blocks can be assumed to be independent from each other. Through a sensitivity study, it was determined that 1000 bootstrap iterations were sufficient to obtain converged uncertainty estimates for the mean. Larger deviations in total mean wind speed can be observed at heights above the rotor tip, which can be attributed to the preference given to the hub height and rotor disc area through weights in the optimization problem. The PDBL simulations have the largest error when comparing the mean veer

between the LES and LiDAR measurements, as by definition, the PDBL simulations have zero veer and are hence incapable of representing a veered flow. While the CNBL simulations do include veer, the TotalControl precursor database was not designed to cover large veer conditions, thus still leading to errors when compared to the veer in measurement data. Nevertheless, the absolute error never exceeds more than 7 degrees over the rotor area for any of the cases at a given range gate. Good comparison is also seen between the measured and simulated turbulence intensity, with the maximum error never exceeding 1%.

## 4.3    Sources of error

The developed optimization framework is therefore able to successfully recreate the inflow conditions at the Lillgrund wind farm for the chosen time periods, without significant errors in the mean inflow velocity across the rotor area, as evident by Figure 9. However, a few shortcomings of the optimization framework must be addressed. Atmospheric stability and stratification can play a significant role in the performance of a wind farm by affecting wake recovery and ambient turbulent intensity

(Magnusson and Smedman, 1994). Additionally, due to the close land proximity of the Lillgrund wind farm, the atmospheric conditions at the farm can be significantly affected by a land–sea system as a combination of marine and land boundary layers. The current methodology does not take the meso-scale effects at the Lillgrund site into account, instead relying on a database comprising of marine pressure driven and neutral boundary layers to match the inflow conditions. This is due to the lack of atmospheric stability and temperature measurements at the Lillgrund site during the measurement campaign. If the data

was available, the inflow database used could have been modified before the transformation procedure to consist of precursor boundary layers which best represent the stability conditions at the wind farm site. In its current form, the considered cases may be biased as they are chosen simply based on minimum error in the distance metrics and do not account for stability or land–sea effects. Additionally, without the inclusion of appropriate weights or additional distance metrics to penalize wind veer error, the current procedure may have been biased to overemphasize mean speed error as compared to wind veer error. The

error in the LiDAR and LES velocity profiles above the rotor area could also affect the vertical transport into the wind turbine array atmospheric boundary layer (WTABL), which may affect the overall wind farm performance (Calaf et al., 2010).

**Table 4.** Summary of the general domain parameters

| | | |
|---|---|---|
| Domain size | $L_x \times L_y \times L_z$ | $16 \times 16 \times 1.5 \ \text{km}^3$ |
| Grid | $N_x \times N_y \times N_z$ | $1200 \times 1200 \times 225$ |
| Resolution | $\Delta_x \times \Delta_y \times \Delta_z$ | $13.33 \times 13.33 \times 6.66 \ \text{m}^3$ |
| Wind-farm spin-up time | $T_{spin}$ | $1800 \times \Delta t_{LES}$ |
| Simulation time | $T$ | $9000 \times \Delta t_{LES}$ |
| Structural time step | $\Delta t_{MB}$ | $0.02 \times \Delta t_{LES}$ |

## 5 Large Eddy Simulations of the Lillgrund wind farm

### 5.1 Numerical setup

The simulation domain in SP-Wind has a size of $16 \times 16 \times 1.5 \ \text{km}^3$ in the stream-wise, span-wise, and vertical directions

respectively. The grid resolution is $13.33 \times 13.33 \times 6.66 \ \text{m}^3$, resulting in a computational grid of $1200 \times 1200 \times 225 = 324 \times 10^6$ grid-points. The choice of domain size is restricted by the one used for the precursor data sets, which was initially designed for simulations with the much larger TotalControl reference wind farm to avoid blockage effects (Sood et al., 2020b). Wind-farm simulations in SP-Wind are performed in a sequence of steps via the concurrent precursor framework (Munters et al., 2016). First, the inflows from the previously generated TotalControl precursor database are made to advance in time in a domain

without wind turbines, called the precursor domain. Concurrently, the flow is transformed using the identified transformation parameters through the optimization framework to obtain the five cases identified in Table 2 and fed into a second domain which contains wind turbines as body forces $\mathbf{F_R}$ through a fringe region. For each of the five cases, the Lillgrund wind-farm is rotated to simulate different wind directions. The flow is allowed to pass through the wind farm for 1800 time steps to account for start-up transients, after which data is collected for evaluating the performance of the farm for 9000 time steps. While the

original precursor inflow database have a LES time step $\Delta t_{LES} = 0.5\,\text{s}$, the time-step of the wind-farm simulations is altered due to the scaling of the velocity field, as per equation 6. The multibody aeroelastic computations are performed with a smaller time step of $\Delta t_{MB}$, for a finer resolution of the structural loading. The general domain and time parameters of the simulations are summarized in Table 4.

### 5.2 Time averaged flow fields

Time-averaged hub height flow fields for all the selected validation cases are shown in Figure 10. It can be seen that out of all the cases, $CNk4_1$ has the highest inflow velocity and $PDk_3$ has the lowest, in accordance with results shown in Figure 9. The different wind directions spanning the five cases lead to different operation states of the same turbines within the Lillgrund wind farm, due to changes in available hub height wind speed, but also due to individual turbines operating in a waked or

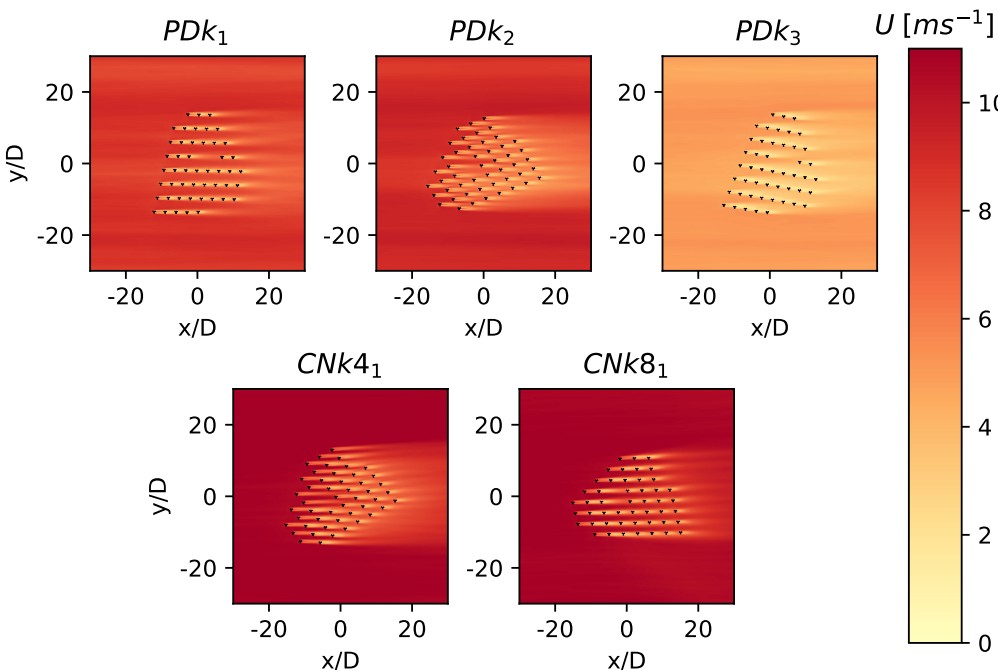

**Figure 10.** Time averaged stream-wise hub height velocity for all selected cases. Different wind directions are realized by rotating the entire wind-farm in the simulation domain and feeding the inflow velocity in the horizontal-x direction.

un-waked condition as per the orientation of the upstream turbines. For instance, the cases $PDk_1$ and $CNk8_1$ with wind directions 119° and 222° respectively, have a larger number of turbines operating under fully waked condition compared to the other three cases. This leads to a data set which allows us to evaluate the performance of Lillgrund turbines when subjected to varying operating conditions.

### 5.3 Performance comparison

Comparison of the mean wind farm power output obtained from SP-Wind against the field measurements from Lillgrund is presented in Figure 11. From Figure 11 (b), it can be seen that the total farm power production in LES for 3 out of 5 cases are in good agreement with the measurement data, with the errors within 95% confidence intervals of each other. While significant errors can be seen for the cases $PDk_3$ and $CNk4_1$, the relative power error for the CNBL case is about 4.2% which is still quite low. The distribution of individual turbine power production for the case $PDk_1$ is shown in Figure 12, including uncertainty estimates again determined by the moving bootstrap method. From Figures 11 (a) and 12 it can be seen that individual turbine power trends across the farm are captured well, where similar trends are observed for power production peaks and valleys for

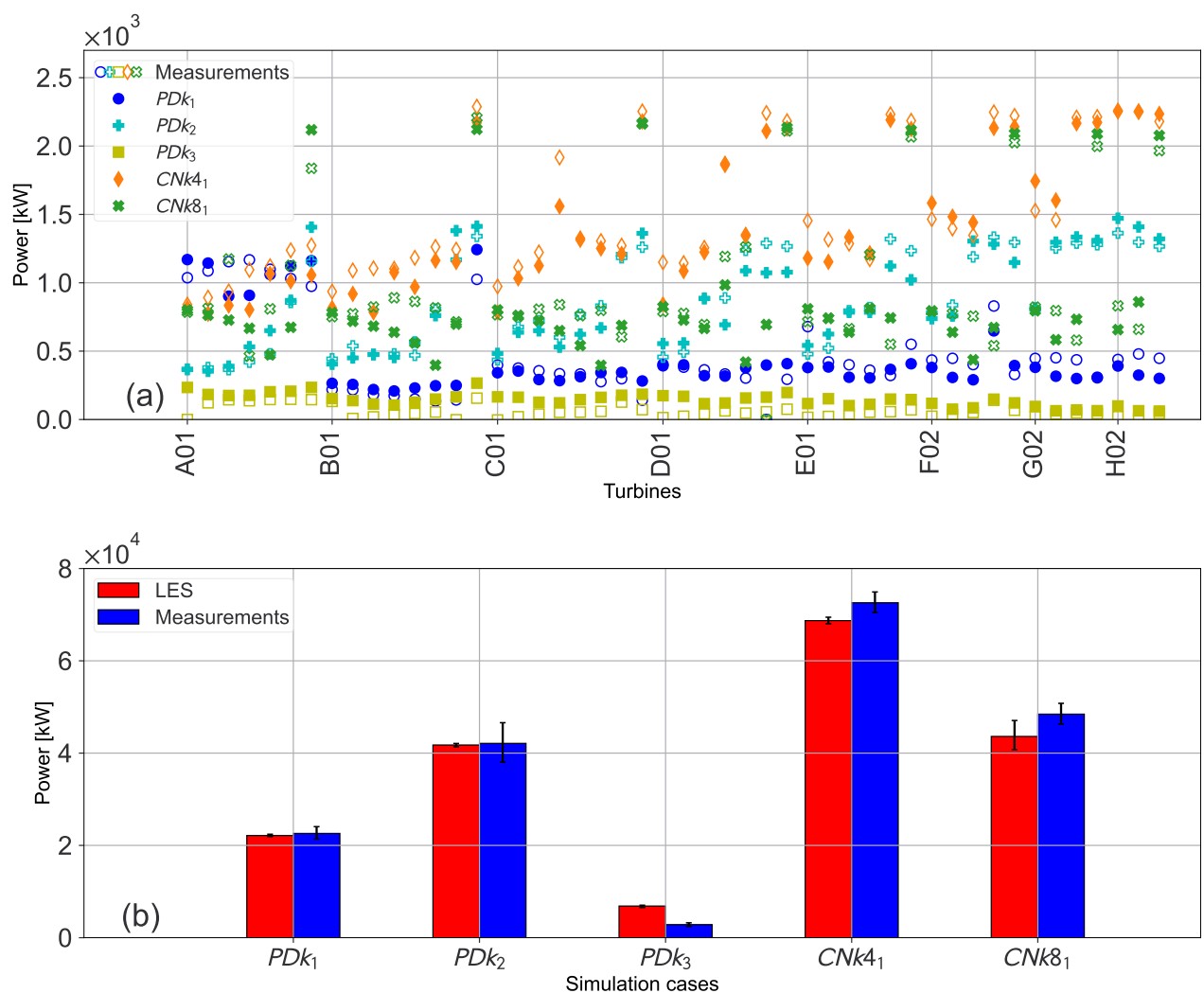

**Figure 11.** Comparison of LES time averaged power output of individual turbines (a) and total wind farm power output (b) against field SCADA measurements. Error bars represent 95 percent confidence intervals on the mean and are computed using the block-bootstrap method and time windows of $600\,\mathrm{s}$ and 1000 bootstrap samples.

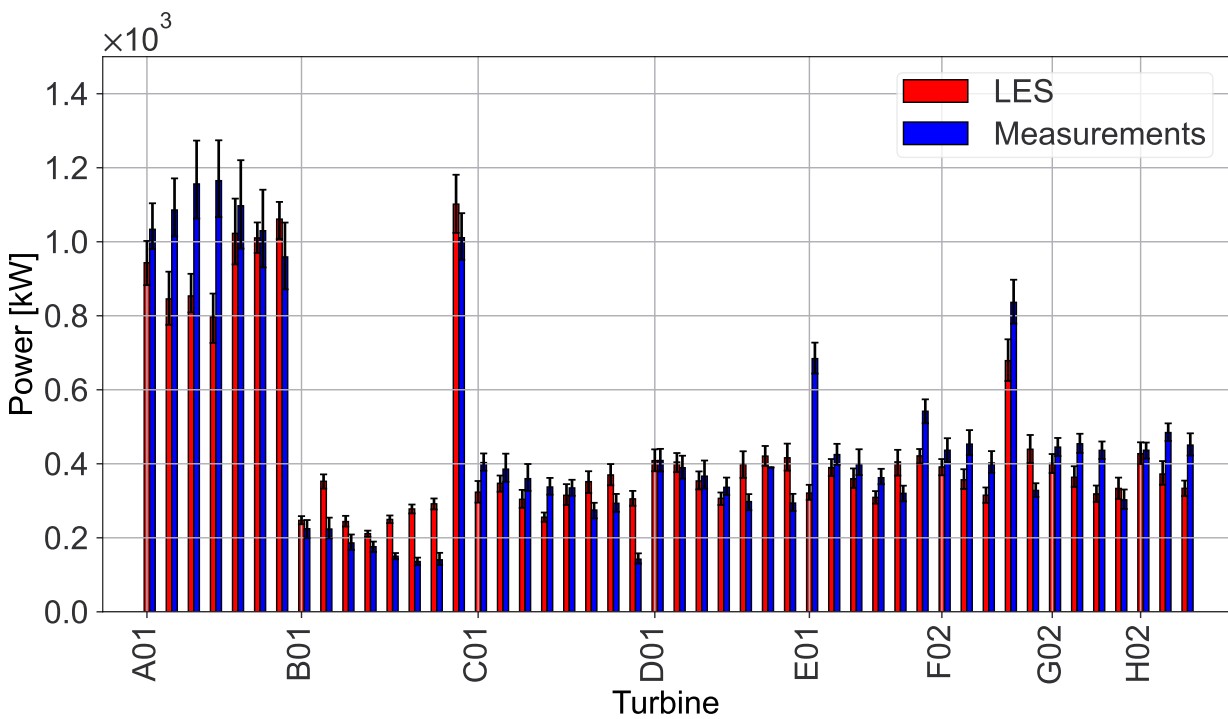

**Figure 12.** Bar plot of turbine power production for the case $PDk_1$. Error bars represent 95 percent confidence intervals on the mean and are computed using the block-bootstrap method and time windows of $600\,\mathrm{s}$ and 1000 bootstrap samples.

**Table 5.** Comparison of percentage of turbines with statistically significant errors based on 95% confidence intervals between LES simulations and field measurements.

| Case | Turbines with significant errors [%] |
|---|---|
| $PDk_1$ | 58.3 |
| $PDk_2$ | 16.6 |
| $PDk_3$ | 83.3 |
| $CNk4_1$ | 25.0 |
| $CNk8_1$ | 35.4 |

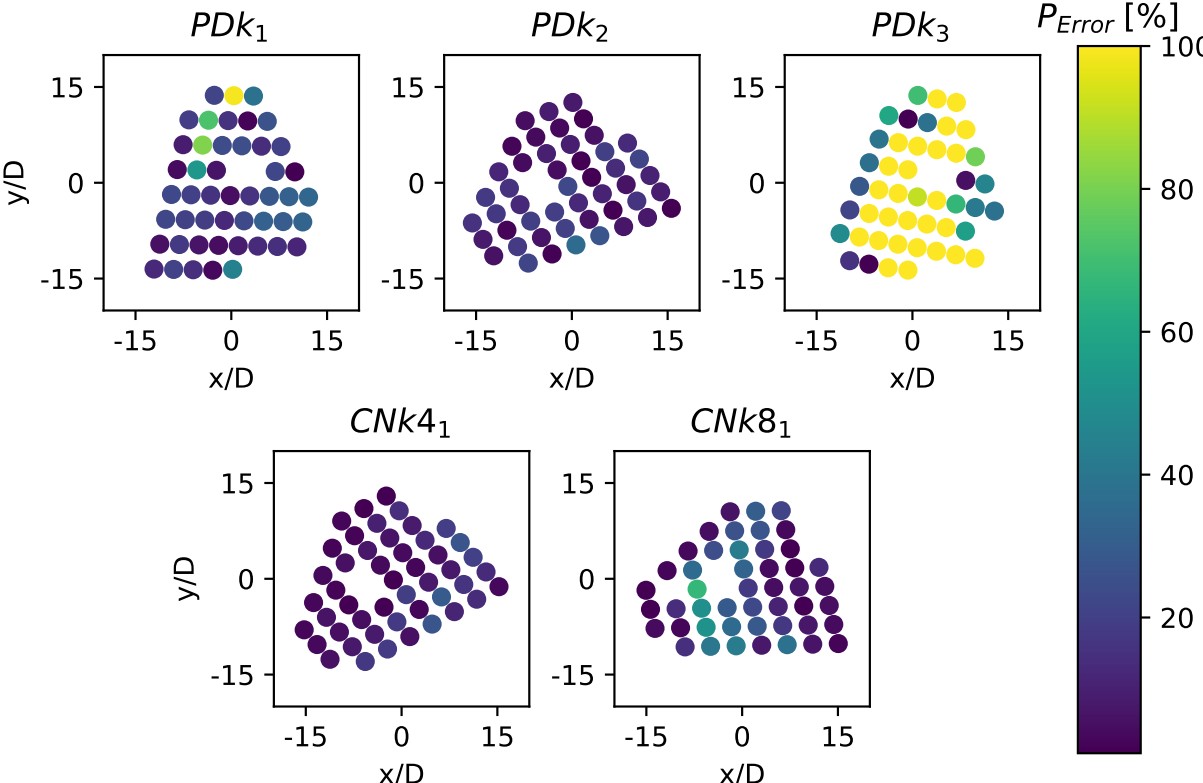

**Figure 13.** Effect of wind direction on relative power error between LES calculations and field SCADA measurements for the 5 simulated cases

un-waked and waked turbines, indicating that on average the wind direction in the LES cases captures the real world field conditions during the time windows. However when comparing individual power production on a turbine level, statistically significant errors are evident for 58% of the turbines for the $PDk_1$ where error bars for the 95% confidence intervals do not overlap when comparing LES predictions and field measurements. The same analysis is performed for all the simulated cases, and the proportion of turbines with statistically significant errors are presented in Table 5. In general, it was observed that the turbines at the front of the farm did not have statistically significant errors, however turbines operating within the wakes of an upstream turbine had higher errors which were significant for wind directions resulting in an aligned configuration.

The effect of the wind direction on the individual turbine power errors are presented in Figure 13. It can be observed that for all the cases, the relative power error is generally low for all the turbines at the front of the farm facing the incoming flow field, further indicating that the replicated flow fields match the inflow conditions at the Lillgrund site. However, turbines within the farm and operating in strong wakes exhibit significantly higher errors, which gradually appear to reduce towards to back of the farm. This behaviour is not visible in the case $PDk_2$ and $CNk_4$, with most of the turbines operating in an un-waked state due to the wind direction. This effect can also be witnessed when observing the proportion of turbines with significant

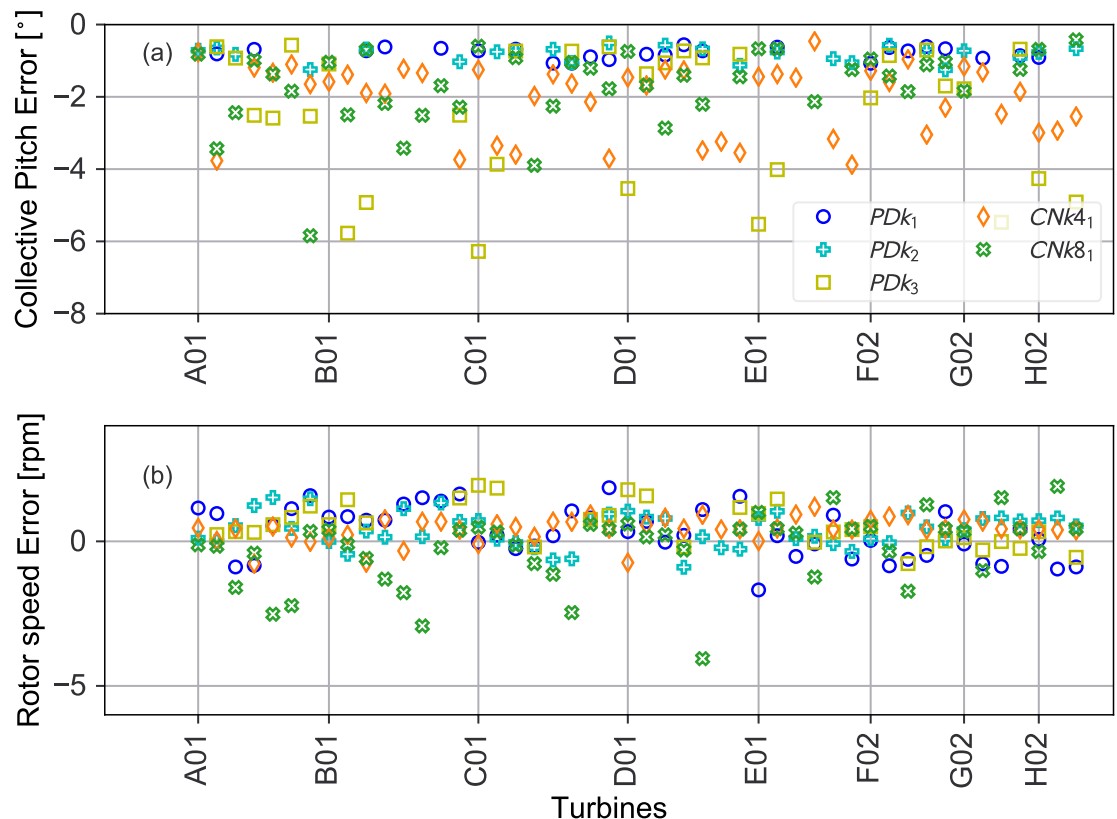

**Figure 14.** Errors between LES time averaged collective pitch (a) and rotational speed (b) of individual turbines and field SCADA measurements. Missing data points correspond to missing data.

errors, as presented in Table 5. The highest errors can be seen for the $PDk_3$ case for almost all the turbines operating inside
the farm, with some of the power errors exceeding 100%. It was observed that while in SP-wind all the turbines for this case
were operational and producing power, most of the turbines in the field were either shut or producing negligible power. To
further investigate the discrepancies in individual turbine power output across the five cases, we investigate the performance
of the implemented controller in the wind farm. Figure 14 shows a comparison between the rotor speed and pitch actuation in
SP-Wind against field measurements for all the turbines. It can be seen that while the performance of the pitch actuation and
rotor speed controller show a good comparison in general across the five cases, larger errors can be seen for a few turbines in
terms of both pitch and rotational speed. In particular, pitch measurements from the wind farm exhibit non-zero pitch angles
for the majority of the turbines, even though they are operating in below rated conditions and should traditionally have zero
pitch, as is the case for the turbines operating in the LES simulations. The highest pitch errors are exhibited by the case $PDk_3$
indicating that instead of operating in Region 2 control regime as expected, the turbines within the farm are operating in a
start-up region with significant pitching action at low wind speeds. This demonstrates the differences between the actual field

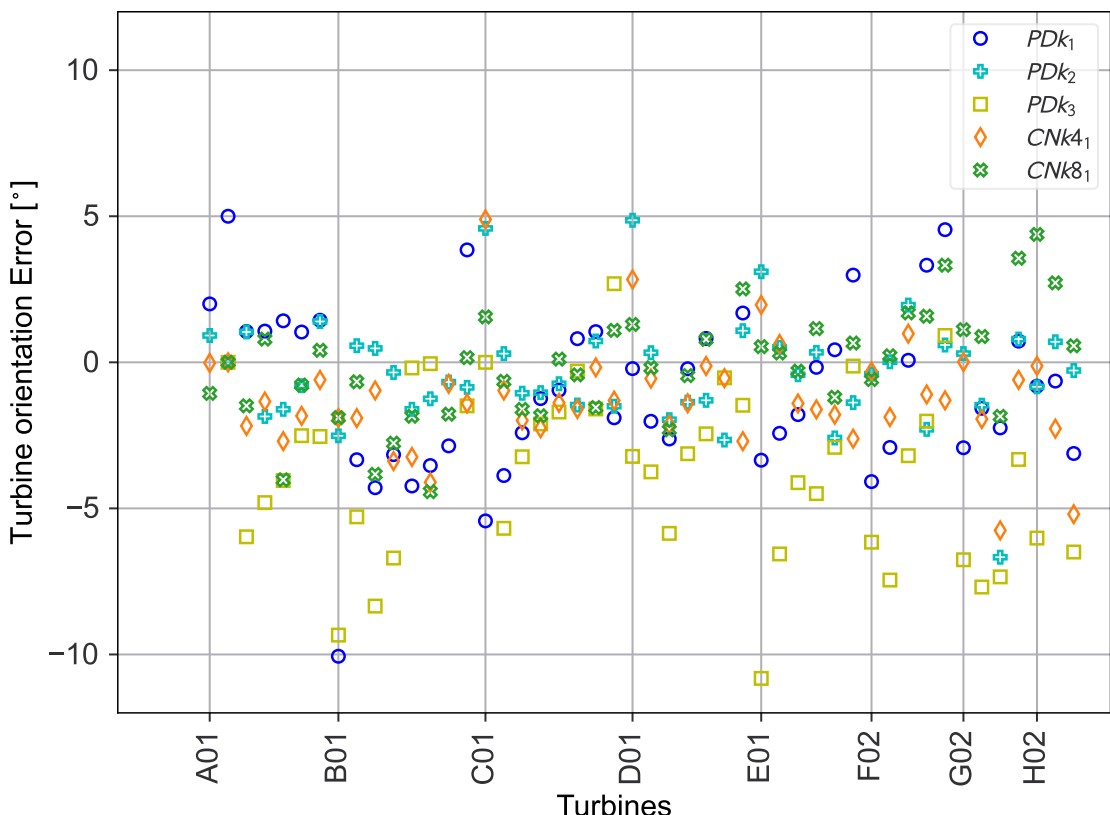

**Figure 15.** Errors between LES time averaged wind direction at individual turbines and field SCADA measurements. All turbines in the LES domain face the wind and have zero yaw misalignment. Missing data points correspond to missing data.

turbine controller and the one implemented in SP-Wind, as detailed information about the controller and all of its modes of operation was not available. Larger pitch angles would lead to reduced power production for the turbines, which can be observed for the $PDk_3$ case with the maximum amount of turbines with non-zero pitch, leading to the larger errors in power production as observed in Figures 11 and 13. Additional controller tuning was attempted by adjusting the gains of the standard torque proportional controller based on the power and rotor speed measurements from the SCADA data. While a reduction in overall farm power error was observed, it was below 1%, which is not significant. Another source of error worth investigating is the yaw misalignment between the simulated cases and the field measurements, which is presented in Figure 15. While all the turbines in the simulation domain of SP-Wind are statically aligned with the mean wind direction without the use of a local yaw angle controller, it can be observed that this was not the case for the field turbines, which dynamically change their orientation in accordance with local wind direction changes within the farm. In fact, the local flow angle for the turbines operating in the LES cases were found to be negligible, indicating that even if a yaw angle controller was incorporated in SP-Wind, it's effect would not have been significant. An analysis of turbine orientation error across the wind farm layout is shown in Figure 16 for

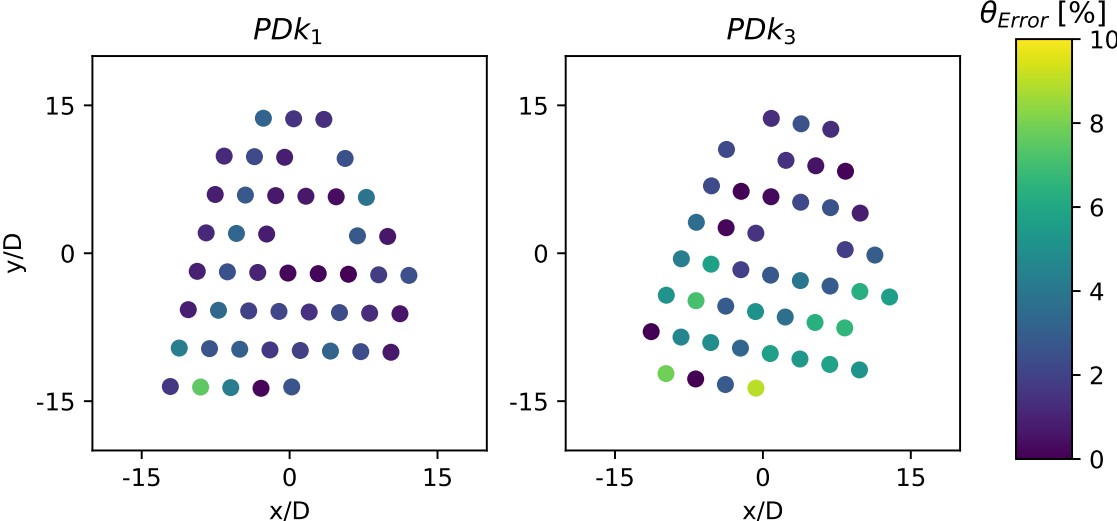

**Figure 16.** Effect of wind direction on relative turbine orientation error between LES calculations and field SCADA measurements for the cases $PDk_1$ and $PDk_3$. Missing data points correspond to missing SCADA data.

cases $PDk_1$ and $PDk_3$ , as they exhibited the highest relative error of the 5 cases. It can be observed that for both the cases, which also have a similar mean wind direction, there is a gradient in the turbine orientation error which increases towards the bottom of the farm. This indicates a gradient in the wind direction across the farm potentially due to meso-scale effects, which the turbine yaw controllers are reacting to. As this gradient is not represented in the LES, we end up with higher turbine orientation errors for a sub-set of the wind turbines.

From the previous analysis, it can be concluded that multiple factors contribute to the errors in performance. It was observed that turbines operating in a strongly waked state exhibit the highest power errors across all the cases. This indicates that the wakes in the LES environment do not recover at the same rate as they do in the field, and is further confirmed by a wake analysis in the following section. As previously remarked in Section 4.3, this could be due to the fact that the optimization framework used to match the inflow conditions at the Lillgrund site did not account for atmospheric stability and meso-scale effects. The larger errors in the mean velocity profile above the rotor area could also be affecting the wake recovery due to vertical transport, and the grid resolution may also be a contributing factor. An analysis of the turbine orientation error across the farm indicated gradients in wind direction in the field which were not recreated in the simulation environment, which could lead to differences in turbine wake overlaps and hence power production errors across the farm. Finally, differences in controller operation in different regimes due to a lack of detailed information about the field controllers, in particular excessive pitching at low wind speeds, can have a significant effect on the observed errors.

Comparison of the mean flap-wise blade root bending moments for five of the 48 turbines from the Lillgrund wind-farm for which loading data was available is shown in Figures 17 and 18 for the PDBL and CNBL simulations, respectively. To

determine the effect of fatigue, we use the damage equivalent loads (DELs) to compare the load histories of the same turbines across the LES and field measurement data. DEL is computed using the Palmgren–Miner rule and the Wöhler equation to account for accumulating fatigue damage caused to the wind turbine components by the fluctuating structural loads (Sutherland, 1999). The loads time series are counted and binned into individual cycles using the rainflow-counting algorithm (Socie and Downing, 1982), and for the wind turbine blades the components follow the Wöhler's curve with a slope coefficient equal to 10 (Freebury and Musial, 2000). The moving block bootstrap methodology is again utilized for evaluating the mean flap-wise moment and the corresponding DEL, with block lengths of 10 minutes and 1000 bootstrap iterations. Fatigue analysis was not conducted for the CNBL simulations, as the data was not logged consistently for the turbines in the time periods of these simulations, making the data unfit for rainflow analysis. Both the average blade root flap-wise moments and the corresponding DELs exhibit a good comparison in the trends of the PDBL simulations in Figure 17. For the $PDk_2$ case, while the moments of the un-waked upstream turbines B08, C08 and D08 show lower errors, significant errors are observed for the turbine B06 which is operating in a fully waked state behind turbine C08. For turbine B07, which is operating in a partially waked state within the farm, while the differences in the flapwise moments is not significant, large errors are visible when comparing the flapwise DEL. A similar observation can be made for the CNBL simulations in Figure 18. Turbines B08, C08 and D08 are operating in free stream conditions for both the CNBL simulations, hence exhibit better comparison against field measurements when compared to turbines B06 and B07 which are operating in fully waked conditions. A possible explanation for the larger errors observed in the loads of the waked turbines could be the relatively coarse grid resolution utilized in SP-Wind when looking at the number of cells across the rotor diameter, such that not all relevant turbulent structures upstream turbine wakes that contribute to loads are captured. In the current work, use of a finer grid for the simulations was not feasible, as the computational cost at the current grid resolutions for the wind-farm simulations is already quite high. Additionally, the larger power error for the waked turbines as observed in Figure 13 would result in differences in blade loading.

## 5.4 Wake analysis

Due to equipment failure, data from the wake measuring LiDAR's was unfortunately available for only two cases, $PDk_3$ and $CNk4_1$, from the five selected validation time periods from the measurement campaign. Comparing wake recovery for the B06 turbine in the $CNk4_1$ case in Figure 19, we see good agreement in wake location and recovery downstream from the turbines. It can be observed that while SP-Wind provides an accurate representation of the near wake region behind turbine B06, the far wake region characterized by a downstream distance greater than 4 rotor diameters exhibits higher errors. This can further be observed from the stronger wakes in SP-Wind in the far wake region for turbine C06 in Figure 20, leading to a lower inflow velocity at turbine C05 and hence the underestimation of power production for turbine C05 as observed in Figure 11. Observing the wakes from the B05 turbine from $PDk_3$ in Figure 21, the effect of incorrectly representing the turbine orientation can be seen. While partial waked conditions are observed in SP-Wind for turbines A05, B05 and C05, the LiDAR measurements exhibit a higher wake overlap. This is observable through the wake deficit analysis shown in Figure 21, where while at first for the LES case the wake behind the turbine B05 appears to be aligned with field measurements, larger deviations are visible further downstream from the turbine where the wake shifts to the left when observed from an upstream

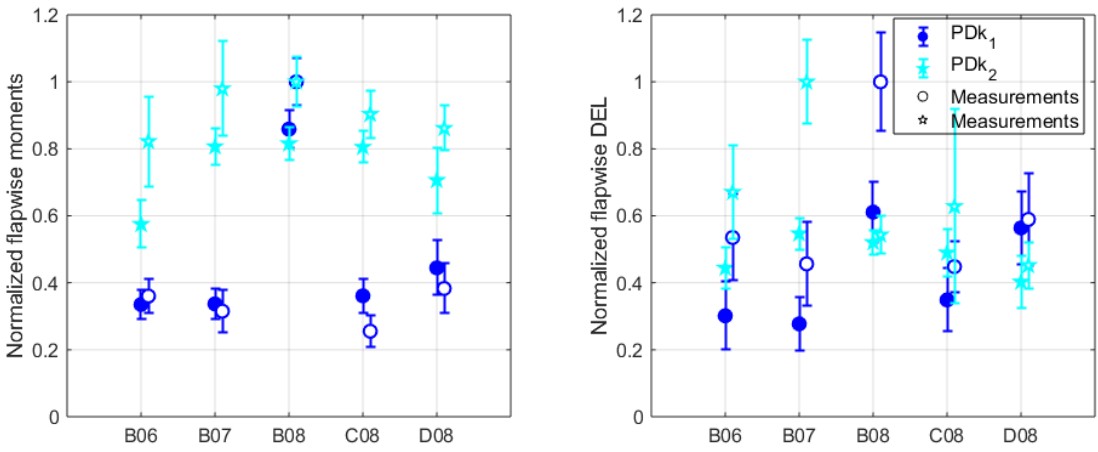

**Figure 17.** Comparison of LES time averaged blade root flap-wise moments (a) and DEL (b) against field measurements. Error bars represent 95% confidence intervals on the mean. Results are normalized by maximum SCADA data for each case.

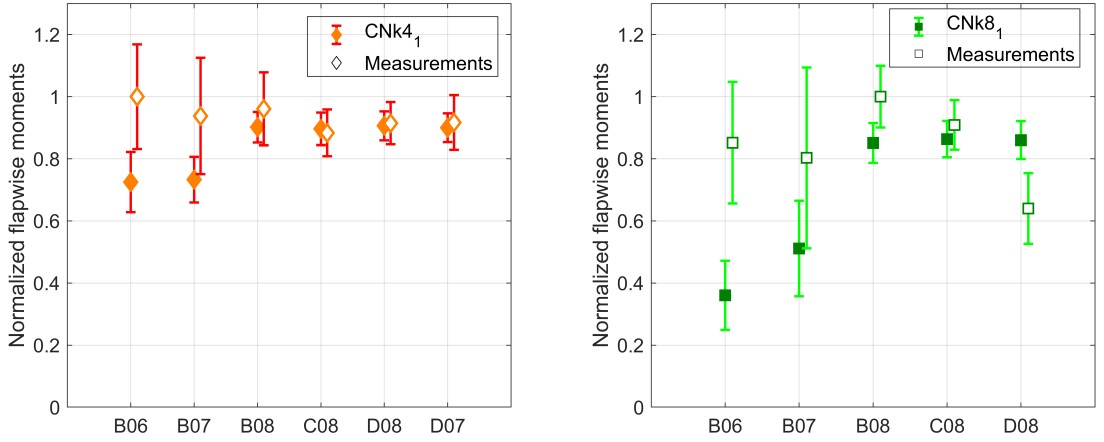

**Figure 18.** Comparison of LES time averaged blade root flap-wise moments for (a) $CNk4_1$ (b) $CNk8_1$ against field measurements. Error bars represent 95% confidence intervals on the mean. Results are normalized by maximum SCADA data for each case.

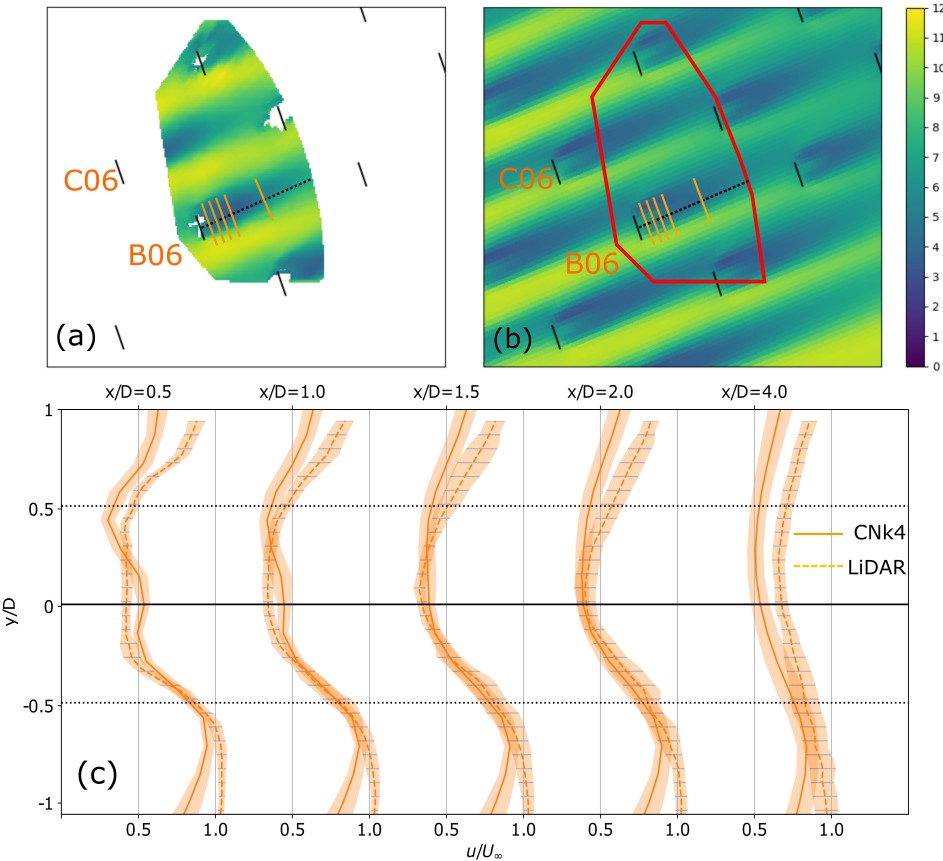

**Figure 19.** Comparison of LES time averaged velocity (b) against LiDAR wake measurements (a) for case $CNk4_1$ and turbine B06.(c) Comparison of velocity wake deficit at downstream locations from turbine B06. Line at y/D=0 is perpendicular to the turbine orientation from SCADA data. Shaded area represents 95% confidence interval on the mean.

point of view. This leads to a larger inflow velocity across the downstream rotor at turbine C05 in LES as compared to LiDAR measurements and hence the greater reported power production in simulations as compared to SCADA data.

## 6 Discussion, conclusions and future work

In this work, a validation study was conducted to compare SP-Wind, a high-fidelity Large Eddy Simulation solver, against field measurements obtained from the Lillgrund offshore wind-farm near the coast of Sweden. To recreate the atmospheric conditions at the Lillgrund site, a framework was developed to create the inflow conditions for the wind farm in the numerical domain by reusing a precursor database through scaling and shifting of the velocity. Thus, the cost intensive step of developing multiple precursor simulations for different atmospheric conditions spanning the duration of the measurement campaign was eliminated. Five time periods from the measurement campaign were selected for simulations in the LES environment.

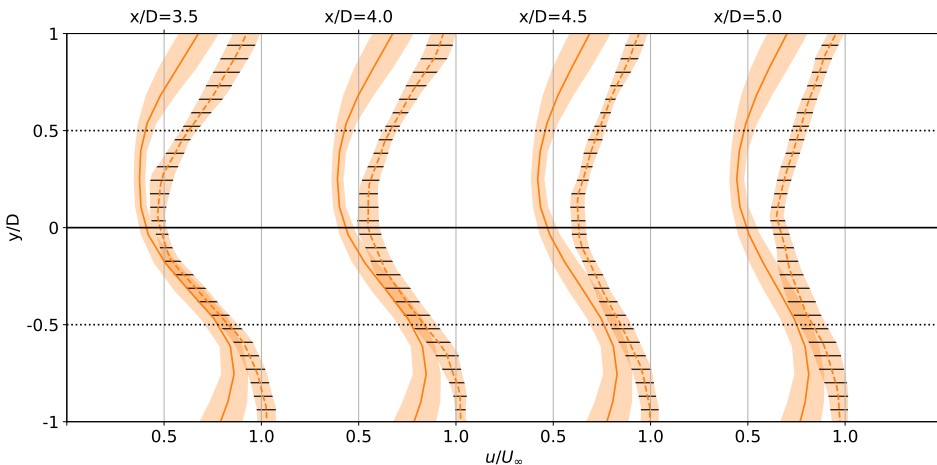

**Figure 20.** Comparison of far wake velocity wake deficit at downstream locations from turbine C06 for the case $CNk4_1$. Line at y/D=0 is perpendicular to the turbine orientation from SCADA data. Shaded area represents 95% confidence interval on the mean.

Upon comparison against field measurements, results from SP-Wind show good comparison in terms of the recreated inflow
velocity field, power produced by individual turbines as well as the entire wind-farm, mean and fatigue blade-root loading
and individual turbine wakes and recovery. However, limitations of the flow solver were exhibited in certain instances. Higher
errors were observed in the performance of turbines operating in a strongly waked state. These errors were evident in both the
turbine power and flapwise load measurements.The discrepancy in the performance of waked turbines could be attributed to
the differences in atmospheric stability between the used precursor boundary layers and the atmospheric boundary layers at
the Lillgrund site, or larger errors in the simulated profiles above the rotor area which could affect vertical transport and wake
recovery behind turbines. Another source of error could be that the grid resolution in the numerical domain may not be fine
enough to capture enough of the relevant smaller turbulent structures in turbine wakes. In the present study, the grid resolution
was limited by the precursor database and computational costs. Additionally, controller mismatch due to lack of information of
the field controllers also lead to discrepancies in the produced power. Particularly, higher errors were evident in a case which
was operating closer to the cut-in wind speed of the turbines located at the Lillgrund site. Nevertheless, the results from the
validation study are promising, proving the capability of a high-fidelity numerical solver to represent on-field conditions and
performance output of a large wind farm.

The analysis in this work thus highlights certain areas in wind farm LES which still require further research to make accurate
performance predictions for offshore wind farms. In particular, lack of prior knowledge of meso-scale effects, field turbine
control logic, and insufficiently fine grid resolution are highlighted as significant actors which can lead to errors in predictions.
Therefore, future work should focus on improving on these effects, in particular for situations with aligned wind turbines.
Further investigation into the role of atmospheric stability and meso-scale effects at the Lillgrund site should also be investi-
gated to determine its role in wake recovery. This can be achieved by expanding the precursor database with finer resolution

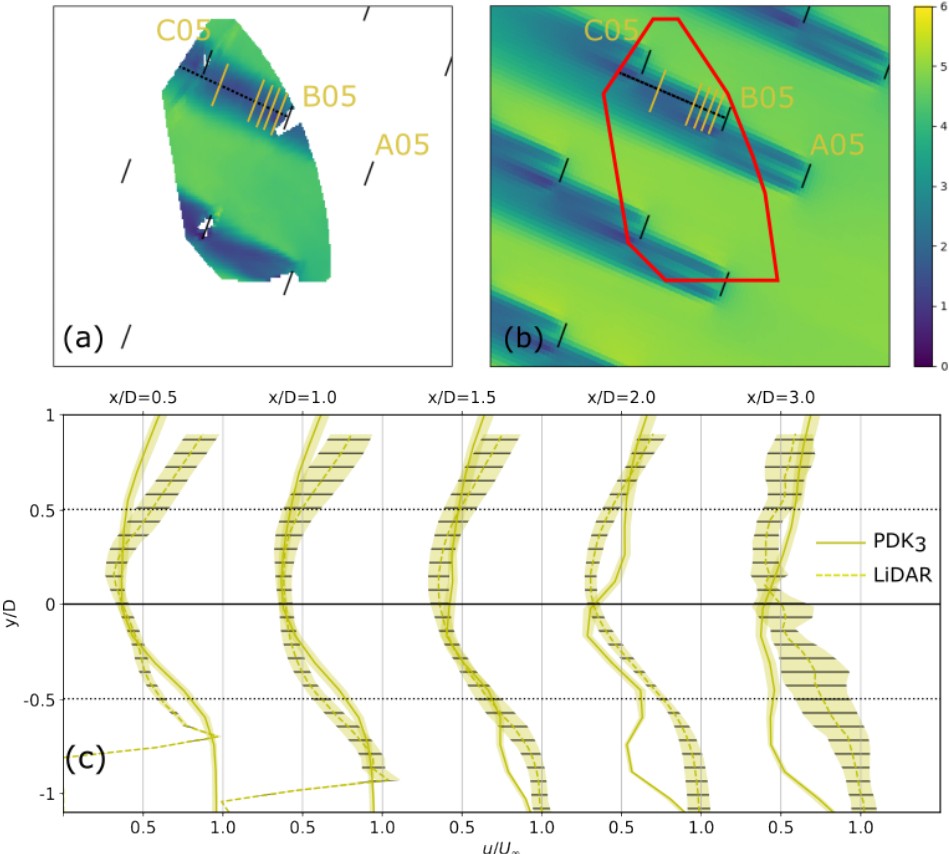

**Figure 21.** Comparison of LES time averaged velocity (b) against LiDAR wake measurements (a) for case $PDk_3$ and turbine B05. (c) Comparison of velocity wake deficit at downstream locations from turbine B05. LiDAR data at x/D=4.0 was not available. Line at y/D=0 is perpendicular to the turbine orientation from SCADA data. Shaded area represents 95% confidence interval on the mean.

data sets, and data sets covering more atmospheric conditions. A closer co-operation with the turbine manufacturers to obtain a detailed controller description for the low wind speed zones and local yaw control dynamics would also be beneficial to improve modelling performance. Having exhibited the capability of the numerical solver in representing normal wind-farm operation, validation studies could also be conducted to evaluate the effect of coordinated wind-farm control strategies, such as wake steering and induction control, to improve wind-farm performance.

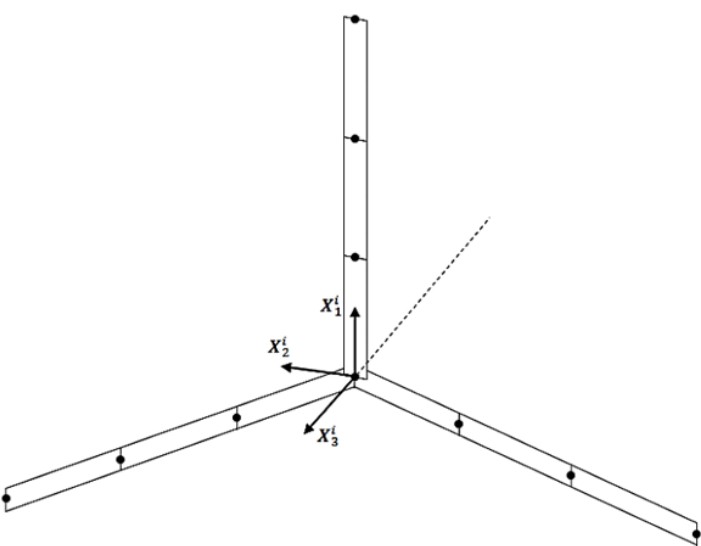

**Figure A1.** Finite element beam representation of the wind turbine blades.

## Appendix A: Multibody model

### A1   Coordinate system

The rotor of the wind turbine is considered as one single body with each blade modeled using a number of interconnected beam elements, as can be seen in Figure A1. The origin of the body reference $X_1^i X_2^i X_3^i$ is located at the center of the turbine rotor and coincides with the root nodes of the finite element beam representations of the respective blades. The $X_1^i$ axis is directed along the length of the first blade, while the axis $X_3^i$ is directed along the axis of rotation of the rotor, in the upwind sense.

The location of the origin of the body reference with respect to the global coordinate system $X_1 X_2 X_3$ (not shown in Fig. 1) is denoted by the vector of Cartesian coordinates $R^i$, while the orientation of the body reference w.r.t. the global coordinate system is denoted by the vector of rotational coordinates $\theta^i$ . Only rotation along the turbine's main axis is assumed to contribute dynamically to the behavior of the turbine; while tilting and yawing motions are taken into account quasi-statically. Finally, the elastic deformations of the rotor are described by the vector of elastic coordinates $q_f^i$ , which describe the displacements of

the finite element nodes and the local derivatives thereof. Summarizing, the configuration of the turbine rotor can be described by the following vector of generalized coordinates:

$$q^i = \left[ R^{iT} \theta^{iT} q_f^{iT} \right]^T \tag{A1}$$

The global position of an arbitrary point on the $j^{th}$ beam element of the $i^{th}$ body of the multibody system can be written as

$$r^{ij} = R^i + \mathbf{A}^i \overline{u}^{ij} \tag{A2}$$

where $\overline{u}^{ij}$ is the displacement vector of the $ij^{th}$ element and $\mathbf{A^i}$ is the transformation matrix of body $i$, which defines the orientation of the body reference with respect to the global reference. The yaw ($\phi$), tilt ($\psi$), rotation ($\theta$) and precone angles ($\gamma$) are used as rotational coordinates about their respective axes, and the resulting transformation matrix cased by these rotations is given by

$$\mathbf{A^i} = \mathbf{R_y^{-1} R_t^{-1} R_r^{-1} R_p^{-1}} \tag{A3}$$

where $\mathbf{R_y}, \mathbf{R_t}, \mathbf{R_r}$ and $\mathbf{R_p}$ are the 3-D yaw, tilt, roll and precone rotation matrices.

## A2 Energy of the turbine rotor

The global velocity of a selected point can be determined by differentiating $r^{ij}$ with respect to time to obtain

$$\dot{\mathbf{r}}^{ij} = \dot{\mathbf{R}}^i + \dot{\mathbf{A}}^i \overline{\mathbf{u}}^{ij} + \mathbf{A}^i \dot{\overline{\mathbf{u}}}^{ij} \tag{A4}$$

which can be simplified by taking into account the assumption of quasi-static variation of yaw, tilt and precone ($\dot{\phi}, \dot{\psi}, \dot{\gamma} = 0$). Thus, the kinetic energy $T^{ij}$ of element $ij$ in the finite element representation of the rotor can be obtained by the following formula

$$T^{ij} = \frac{1}{2} \int_{V^{ij}} \rho^{ij} \dot{\mathbf{r}}^{ij^T} \dot{\mathbf{r}}^{ij} dV^{ij} \tag{A5}$$

where $V^{ij}$ and $\rho^{ij}$ are the mass density and volume of the $ij^{th}$ element, respectively. The total kinetic energy of the turbine rotor can be determined by summing up the kinetic energies of all the elements. Using the floating frame of reference approach (Shabana, 2013), the expression for the elastic potential energy takes a very simple form, since only the elastic (i.e. not the rigid body) displacement of the body contributes to the elastic potential energy. Consequently, the potential energy of the $ij^{th}$ element is given by

$$\Pi^{ij} = \frac{1}{2} {\mathbf{q}_f^i}^T \mathbf{K_{ff}}^{ij} \mathbf{q}_f^i \tag{A6}$$

where $\mathbf{K_{ff}^{ij}}$ is the the element stiffness matrix expressed with respect to the body reference frame.

## A3 Equations of motion

The equations of motion of the turbine rotor are developed from Lagrange's equations for constrained systems. These equations can be written in the following form:

$$\frac{d}{dt} \left( \frac{\partial T^i}{\partial \dot{\mathbf{q}}^i} \right)^T - \left( \frac{\partial T^i}{\partial \mathbf{q}^i} \right)^T + \left( \frac{\partial \Pi^i}{\partial \mathbf{q}^i} \right)^T + \mathbf{C_{q^i}^T} \lambda = \boldsymbol{Q}^i \tag{A7}$$

where $T^i$ and $\Pi^i$ are the kinetic and potential energy of the $i^{th}$ body, $q^i$ is the vector of generalized coordinates of the $i^{th}$ body, and $\boldsymbol{Q^i}$ is the vector of generalized forces associated with the coordinates of the $i^{th}$ body. Furthermore, $\boldsymbol{\lambda}$ is the vector

of Lagrange multipliers and is the $\mathbf{C_q^i}$ constraint Jacobian matrix, defined as

$$\mathbf{C_q^i} = \frac{\partial \boldsymbol{C}}{\partial \boldsymbol{q}^i} = \begin{bmatrix} \frac{\partial C_1}{\partial q_1^i} & \frac{\partial C_1}{\partial q_2^i} & \cdots & \frac{\partial C_1}{\partial q_n^i} \\ \frac{\partial C_2}{\partial q_1^i} & \frac{\partial C_2}{\partial q_2^i} & \cdots & \frac{\partial C_2}{\partial q_n^i} \\ \vdots & \vdots & \ddots & \cdots \\ \frac{\partial C_{n_c}}{\partial q_1^i} & \frac{\partial C_{n_c}}{\partial q_2^i} & \cdots & \frac{\partial C_{n_c}}{\partial q_n^i} \end{bmatrix} \tag{A8}$$

where $\boldsymbol{C} = \boldsymbol{C}(\boldsymbol{q},t) = (C_1 C_2 \ldots C_{n_c})^T$ is the vector of linearly independent constraint functions that satisfy the holonomic constraint equations of the multibody system

$$\boldsymbol{C}(\boldsymbol{q},t) = 0 \tag{A9}$$

After evaluating and expanding the partial derivatives in equation A7 in terms of the mass and stiffness elements of the rotor structure, we can obtain the equation of motion of the multibody structure as

$$\mathbf{M^i}\ddot{\boldsymbol{q}}^i + \mathbf{K^i}\boldsymbol{q}^i + \mathbf{C_q^T}\lambda = \boldsymbol{Q}_e^i + \boldsymbol{Q}_v^i \tag{A10}$$

where $\boldsymbol{Q}_e^i$ is the vector of generalized external forces containing the aerodynamic and gravity forces at each body element and $\boldsymbol{Q}_v^i$ is the quadratic velocity vector of body i, as defined by

$$\boldsymbol{Q}_v^i = \frac{1}{2} \left[ \frac{\partial}{\partial \mathbf{q}^i} \left( \dot{\mathbf{q}}^{iT} \mathbf{M}^i \dot{\mathbf{q}}^i \right) \right]^T - \dot{\mathbf{M}}^i \dot{\mathbf{q}}^i \tag{A11}$$

## Appendix B: Coupling of turbine model with flow solver

During each LES time step, the blades sweep a sector area where the loads and their dynamic response are evaluated in a two-way Fluid-Structure Interaction (FSI) manner. Loads acting on the turbine and tower structure lead to deformations, which are evaluated using equation 3, and the subsequent loads are then computed on the structure's deformed positions, before being added to the flow equations 1 as body force terms $\bar{\mathbf{F}}$. Before being added to the flow equations, the body forces of are processed by spatial and time filtering, as detailed in the following subsections.

### B1  Spatial filter of rotor-swept forces

First, the unsteady forces $\mathbf{F}(\hat{\mathbf{u}}, \mathbf{q})$ are smeared out in the surrounding LES mesh nodes by taking their convolution with a Gaussian kernel $G_n$, resulting in the spatially filtered forces $\hat{F}(x)$:

$$\hat{F}(\mathbf{x}) = \sum_{n=1}^{N_t} \sum_{j=1}^{N_b=3} \int_0^R F(\hat{u},q,r) G_n(\|\mathbf{x} - r\mathbf{e_j}\|) dr \tag{B1}$$

where, $N_t$ is the number of turbines, $N_b$ the number of blades, and $\|\mathbf{x} - r\mathbf{e_j}\|$ is the Euclidean distance between the LES grid point and the deflected actuator line point, accounting for structural deformations.

## B2    Time filter of rotor-swept forces

The body forces $\bar{\mathbf{F}}$ of the Navier-Stokes equations are then calculated by time-filtering the Gaussian-filtered forces $\hat{F}$ through a first-order low-pass filter, which gives more weight to the last sub-iterations (see Figure 4d). The time-filter is given as follows:

$$\frac{d\bar{F}^i}{dt} = \frac{1}{\tau_f} \left[ \hat{F} - \bar{F}^i \right] \tag{B2}$$

where, $\bar{F}^i$ is the time-filtered force at Runge-Kutta stage $i$, $\hat{F}$ are the spatially filtered forces through the sub-cycles, and $\tau_f$ is the time filter constant. The filter constant $\tau_f$ defines the effective sector angle, which is chosen to be equal to the LES time step. Equation B2 is integrated during the sub-iterations of the multibody solver using an implicit Euler scheme. The choice of using a first-order filter is justified by its simplicity, however future work could focus on studying the influence of higher order filters.

*Author contributions.* IS and JM jointly set up the concept and objectives of the current work. IS performed code implementations in the LES code SP-Wind and carried out the wind farm simulations. AV and BB developed and implemented the multibody framework and turbine model. ES conducted the measurement campaign and data analysis under the guidance of GL. IS selected the validation cases with support from JM. IS performed the data analysis and data visualization. IS and JM wrote the manuscript with contributions and revisions from ES, AV and GL.

*Code and data availability.* The SP-Wind flow solver is a proprietary software of KU Leuven. The TotalControl inflow database used in this work to recreate the inflow conditions at the Lillgrund wind farm is publicly available on zenodo (Munters et al., 2019a, b, c, d). Time averaged LES inflow, SCADA and inflow LIDAR measurements are available online at https://doi.org/10.5281/zenodo.7358841. Rest of the raw data of the simulation results can be obtained by contacting the corresponding author.

*Competing interests.* The authors declare that they have no conflict of interest.

*Acknowledgements.* The authors have received funding from the European Unions Horizon 2020 program (TotalControl, grant no. 727680). The computational resources and services used in this work were provided by the VSC (Flemish Supercomputer Center), funded by the Research Foundation Flanders (FWO) and the Flemish Government department EWI. The authors also acknowledge DTU, Siemens Gamesa and Vattenfall for providing the LiDAR, loads and SCADA data, respectively.

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
