# Peer review of "Comparison of Large Eddy Simulations against measurements from the Lillgrund offshore wind farm"

_Wind Energy Science, 2021_

## Referee Comment (RC2)

Review of manuscript: WES-2021-153
Title: Comparison of Large Eddy Simulations against measurements from the Lillgrund offshore wind farm
Authors: Sood et al.

The submitted paper focuses on a detailed comparison between large eddy simulations (LES) of the Lillgrund offshore wind farm and experimental LiDAR and SCADA data. Overall, I am impressed with the technical challenge the authors have approached. Further detailed comparisons between LES and real wind farm measurements are needed in the literature to further establish confidence in farm models, and to identify areas which require modeling improvements. The authors have provided the appropriate detail regarding their advanced LES solver and wind turbine representation, which includes turbine rotation and loads. Finally, the field data itself is comprehensive and compelling.

However, I do have several comments and suggestions which I encourage the authors to consider in a revision. Principally, while significant attention is placed on the setup of the study (~20 pages including Appendices), there is less attention to detail in the analysis of the results, the uncertainties of the analysis, conclusions, and future work (~8 pages which are mostly figures, see point comments below).

I suggest that the authors add a new section which outlines the uncertainties, gaps in knowledge/modeling, and highlights the future work that this study motivates. There appear to be many areas which need to be improved before sufficient accuracy is obtained in LES wind farm modeling, especially in ABL representation. Several figures can also be improved in their visibility.

In the abstract, the authors state that 'good' agreement is obtained between LES and field measurements for power, loading, and wake recovery. I suggest to include quantitative descriptions of the error and to eliminate the qualitative descriptors, since different readers may have different take-aways from the results (e.g. looking at Figure 11, there appear to be statistically significant differences in the power between LES and measurements for around half of the turbines in the farm, which I would argue is not a good agreement). I also appreciate that the authors refer to some errors in the abstract, and the need for improved controls modeling and a finer grid.

Finally, especially given the preliminary nature of the comparison between model results and field LiDAR data, I encourage the authors to consider publishing the wind farm LiDAR and SCADA measurement dataset along with their paper. This would be valuable to the community.

Point comments:

1. Abstract, Line 10: What is the optimization framework mentioned here? To this point, optimization hasn't been mentioned in the abstract.
2. Abstract: Quantitative results in the abstract would be useful to add.
3. Line 46: Typographical error.
4. Figure 1:

     a.   Overall, I found this figure to be a bit confusing. Consider labeling elements within the figure, in addition to a discussion in the caption. More information could also be added to the caption (e.g. defining the red and blue arcs, which are not currently defined).

     b.   Define Vara, Levante, Sternn.

     c.   The turbine number labels for the turbines which have a mounted LiDAR are missing.

     d.   Consider adding an upwind facing arrow to the magenta inflow centerline. Please also define it in the caption.

     e.   Why are the red and blue scan areas chosen not symmetric?

     f.   More details are required to explain the "transect lines". The Figure mentions plural "transect lines" but I see only one line?

5.   Line 78: Was the turbine nacelle direction only of 'Vara' used to characterize the true inflow direction or was averaging over multiple turbines performed?

6.   Line 81: How high above sea level was the LiDAR mounted?

7.   Section 2.2.2 was difficult to follow. What are the "three transects"? Is the third transect the upwind inflow LiDAR transect? Why was 10-minute averaging selected? Is this synchronized with the SCADA data?

8.   Line 96: Did the processing to result in uniform 0.5 Hz sampling involve upsampling or downsampling?

9.   Line 99: Why was the nacelle calibration performed using wake losses (assuming that the wake advects exactly with the mean wind) for the SCADA data but it was performed using nacelle direction measurements for the inflow LiDAR? What magnitude of uncertainty do the authors estimate is incorporated through these correction methods?

10.  Line 99: Typographical error.

11.  Line 126: *"The effect of the sea surface is included using a wall-stress model [...]"*
At first, I found this to be confusing because as written it seems to imply that the effect of waves are considered (but that is just my interpretation), but this is a standard wall model. Consider rephrasing the sentence?

12.  Sections 3.2 and 3.3: These sections (along with Appendices) are clear and detailed. Have the authors previously compared this AASM model to ALM, ADM, etc? There are a growing number of actuator turbine representations (e.g. actuator surface). It would be helpful to add a sentence or two to concisely summarize pros and cons of the selected AASM methodology.

13.  Figure 4: I think this figure is really great, very clear. Why was a first-order time-filter selected? Has this decision been validated in previous papers/theses by the authors for the AASM? If so, please provide references and justification. Also, consider defining variables in the figure caption.

14.  Figure 5: Are the deviations in this figure the result of a difference in the induction given the coarse grid resolution in SP-Wind?

15.  Figures 6 and 7: I presume that the individual data points shown are not the full resolution of the precursor LES cases (e.g. CNk8 has two points in the boundary layer). Please clarify.

16.  Section 4.1:

     a.   Does $u^t$ represent the velocity vector (it is not bolded and no vector identification is included) or wind speed? If it is wind speed, is the spanwise velocity (v) neglected from

this analysis? If it is a vector, are there unique values of $u^{*,t}$ and $z_0^{\,t}$ for each of the streamwise and spanwise velocities?

    b. How do the authors replicate the wind veer from the LiDAR? This is not included in Eqs. (5) or (6).

    c. Also, e1 is not defined in Eq. (5).

    d. This discussion and the references focus on surface layer similarity. Cases with geostrophic forcing at a finite Rossby number are also considered. Matching the friction velocity can lead to large errors in the geostrophic wind speed (as seen in Figure 8).

17. Line 225: What does it mean for the range gate to span the hub height, as the hub height is one location.

18. Table 3: What is the uncertainty associated with the TI estimation for both LiDAR and LES?

19. Figure 8 is messy with the inset figures overlapping the outer figures. Please consider reformatting. Also, how were the 95% CIs estimated for both LiDAR and LES? Please add this description to the text body.

20. Section 4.2: Overall, I find this section and the outlined approach interesting. I recognize the technical challenges of what the authors are attempting, given fundamental inconsistencies between the CN or PD LES cases and the true ABL LiDAR measurements. Primarily, fitting neutral or PD ABL profiles to non-neutral ABLs may bias the results. So, I suggest the authors add an additional paragraph, addressing the following:

    a. Discussion of the stability during the LiDAR measurement time periods.

    b. Discussion of how the optimization process which finds the minimum error compared to conventionally neutral or pressure driven boundary layers (with no stratification) may bias the considered cases.

    c. Given the poor agreement in the wind veer, discuss how the optimization process may have biased the results to over-emphasize wind speed compared to wind direction.

    d. Discussion of how the pronounced disagreement between LiDAR and LES above the rotor area may bias the vertical transport into the wind turbine array atmospheric boundary layer (WTABL).

    e. Finally, I find it a bit strange that several of the selected cases had LiDAR or loads measurements missing. Why did the authors not select other cases where all of the data were present (even if the precursor data differed slightly more than in other cases)?

21. Line 286: *"This is in accordance with the errors in the inflow profiles observed [...]"*
The analysis of the power results in this sentence appears weak and selective. First, putting aside the power and looking just at the wind profiles in Figure 8, all of the PD cases have high error in the wind veer, not just PDk3. Second, it is not clear from Figure 8 that the PD cases have lower wind speed error than the CN cases. For example, CNk8 seems to have fairly low wind speed and wind veer error. Cases PDk1 and PDk2 do not appear to have significantly lower wind speed error compared to the LiDAR than CNk8. The authors should revisit this analysis with more attention. I suggest quantitative metrics to support the statements.

22. Figure 10: How sensitive are the bootstrapped CIs to the block length selected in the bootstrapping approach?

23. Figure 11 is challenging to see. Please modify the aspect ratio to be wider and shorter.

24. Line 304: Do the LES turbines align themselves with the wind direction measured locally at each turbine?

25. Figure 13: I believe the y-axis of this plot should be 'Turbine orientation' or 'Nacelle position' rather than wind direction. In the field, the turbine's orientation is not always aligned with the wind direction, depending on the yaw controller dynamics. The local wind direction may also vary within the farm. Also, why are the turbine position errors high for all turbines in PD1? Does this indicate that the calculated mean wind direction was not correct?

26. Line 310: *"This indicates [...]"*
This sentence seems to contradict the previous sentence.

27. Figure 14: The difference between measurements and LES for the star symbols is not clear in these figures. The open star looks almost filled.

28. Line 323: *"For the PDk2 case [...]"*
Again, I encourage the authors to be more precise in their analysis. For PDk2, with the errorbars in Figure 14 representing 95% CIs, there are statistically significant errors between LES and LiDAR for B08 and B06, while the differences between LES and LIDAR for B07 are not statistically significant. This differs from the present analysis discussion in the paper.

29. Figures 16 and 17: Why is the far wake comparison omitted from these plots? Please include. They are mentioned on line 335, so presumably the data is there. If they were selectively omitted due to poor agreement between LES and LiDAR, that is even more reason to include them in the figures.

30. Line 337: *"Observing the wakes [...]"*
I was not able to discern the turbine orientation story from the LiDAR results in figure 17(a). Could the authors rephrase this discussion to be more clear? Looking at figures 17(a) and 17(b), how is it clear that in LES there is partial wake while in LiDAR there is not, since only C05 can be seen (and it does appear to be in a partial wake).

---

## Author Comment (AC2)

**Response to reviewer 2**

The submitted paper focuses on a detailed comparison between large eddy simulations (LES) of the Lillgrund offshore wind farm and experimental LiDAR and SCADA data. Overall, I am impressed with the technical challenge the authors have approached. Further detailed comparisons between LES and real wind farm measurements are needed in the literature to further establish confidence in farm models, and to identify areas which require modeling improvements. The authors have provided the appropriate detail regarding their advanced LES solver and wind turbine representation, which includes turbine rotation and loads. Finally, the field data itself is comprehensive and compelling.

However, I do have several comments and suggestions which I encourage the authors to consider in a revision. Principally, while significant attention is placed on the setup of the study (~20 pages including Appendices), there is less attention to detail in the analysis of the results, the uncertainties of the analysis, conclusions, and future work (~8 pages which are mostly figures, see point comments below). I suggest that the authors add a new section which outlines the uncertainties, gaps in knowledge/modeling, and highlights the future work that this study motivates. There appear to be many areas which need to be improved before sufficient accuracy is obtained in LES wind farm modeling, especially in ABL representation. Several figures can also be improved in their visibility.

We thank the reviewer for the comments and suggestions which we believe have significantly improved the quality of the manuscript. As per the suggestion, we have expanded upon the results and the conclusions of the study and included discussions regarding the uncertainties of the analysis and sources of errors. We have also updated the figures mentioned based on the suggestions to improve their visibility.

In the abstract, the authors state that 'good' agreement is obtained between LES and field measurements for power, loading, and wake recovery. I suggest to include quantitative descriptions of the error and to eliminate the qualitative descriptors, since different readers may have different take-aways from the results (e.g. looking at Figure 11, there appear to be statistically significant differences in the power between LES and measurements for around half of the turbines in the farm which I would argue is not a good agreement). I also appreciate that the authors refer to some errors in the abstract, and the need for improved controls modeling and a finer grid.

We have no modified the abstract, quantitatively highlighting the differences observed in the analysis. The following lines have now been added

"For four out of the five simulated cases, the total wind farm power error was found to be below 5%. Nevertheless, statistically significant errors were observed on an individual level for a few turbines deeper in the wind farm operating in wakes across the simulated cases. While the compared flapwise loads in general show a reasonable agreement, errors greater than 200% were also present in some cases. Larger errors in the wake recovery in the far wake region behind the LiDAR installed turbines were also observed."

Finally, especially given the preliminary nature of the comparison between model results and field LiDAR data, I encourage the authors to consider publishing the wind farm LiDAR and SCADA measurement dataset along with their paper. This would be valuable to the community.

We are currently discussing with the project partners who provided the LiDAR, SCADA and loads measurements to determine which data sets can be made publicly available. We will include these on an online repository under the 'data availability' section in the final publication.

**Please find below responses to the reviewers point comments.**

1.      Abstract, Line 10: What is the optimization framework mentioned here? To this point, optimization hasn't been mentioned in the abstract.

This has been corrected and the framework is mentioned earlier in the abstract.

2.      Abstract: Quantitative results in the abstract would be useful to add.

We have modified the abstract and included the information that statistically significant errors are present when comparing the LES against field measurements. We have added quantitative measures for differences in power production and loading in the abstract.

3.      Line 46: Typographical error.

Corrected in text.

4.      Figure 1:

We thank the reviewer for the suggestions, and the following corrections/modifications are implemented to improve the figure.

   a.   Overall, I found this figure to be a bit confusing. Consider labeling elements within the figure, in addition to a discussion in the caption. More information could also be added to the caption (e.g. defining the red and blue arcs, which are not currently defined).

   Added text in the figure caption and modified the figure

   b.   Define Vara, Levante, Sternn.

   Added text in the figure caption and modified the figure

   c.   The turbine number labels for the turbines which have a mounted LiDAR are missing.

   Modified the figure

   d.   Consider adding an upwind facing arrow to the magenta inflow centerline. Please also define it in the caption.

   The magenta inflow centreline indicates the central azimuth angle of the PPI sector scan where the reconstruction result provides wind vectors across. It is not connected to the inflow wind direction to the site.

   e.   Why are the red and blue scan areas chosen not symmetric?

   The following lines have been added in section 2.2.1 to answer this point

   "The wake scan positions (red and blue areas) were determined through an optimization where the LiDAR positions and points along the wake transect were first defined, and a trajectory for eachLLiDAR was generated which matched the position and timing of both LiDARs, along with kinematic controls to control the scan speed, motor acceleration, etc. The scanned areas are mostly symmetric, but not exactly due to the actual installed positions and wake transect point locations chosen.

   f.   More details are required to explain the "transect lines". The Figure mentions plural "transect lines" but I see only one line?\

   Added in figure caption. The wake transect is scanned at 3 heights but this is not visible in the top-down view.

For completeness, the modified caption of the figure to address the above points is mentioned below

"Top-down view of the the Lillgrund site and scan areas for inflow and wake measuring LiDARs installed on turbines B08 "Vara", A07 "Levante" and C07 "Sterenn". Coordinates are relative to the turbine B08 position, spaced in multiples of 5 rotor diameters. Vara performs PPI sector scans which are reconstructed into wind vectors along the magenta inflow centerline. Sterenn and Levante perform coordinated dual-Doppler complex trajectory scans which provide time and space synchronized measurements at 3 heights along the wake transect indicated with black crosses. The overlapping area between the areas scanned in red by Levante and blue by Sterenn are shaded in green. The positions within the green area which lie off the synchronized wake transect line are resolved through time-averaging and applying dual-Doppler retrieval to the 10-minute averaged wind field."

5.      Line 78: Was the turbine nacelle direction only of 'Vara' used to characterize the true inflow direction or was averaging over multiple turbines performed?

Added in text. Only the (calibrated) nacelle direction SCADA from turbine B08 was used to determine the azimuth offset of the inflow lidar, there was no averaging or consensus method used with multiple turbines.

6.      Line 81: How high above sea level was the LiDAR mounted?

Added in text. The scan heads of the lidars were measured at 5m above mean-sea-level using a GNSS sensor with RTK correction. The transition piece height is 3.5 m ASL.

7.      Section 2.2.2 was difficult to follow. What are the "three transects"? Is the third transect the upwind inflow LiDAR transect? Why was 10-minute averaging selected? Is this synchronized with the SCADA data?

Clarified in text. This section refers only to the wake scan strategy. The three wake transects follow the same Northing, Easting (X,Y) positions shown in Figure 1 (the line with black crosses), but are repeated at 3 heights (Z $\in$ {6,70,130m ASL}) which is not visible from the top-down view provided in the figure.

8.      Line 96: Did the processing to result in uniform 0.5 Hz sampling involve upsampling or downsampling?

Added in text. The channels and periods with lower than 0.5 Hz sampling were upsampled (interpolated) to achieve 0.5Hz across all channels/periods.

9.      Line 99: Why was the nacelle calibration performed using wake losses (assuming that the wake advects exactly with the mean wind) for the SCADA data but it was performed using nacelle direction measurements for the inflow LiDAR? What magnitude of uncertainty do the authors estimate is incorporated through these correction methods?

The inflow lidar's azimuth offset was determined using the calibrated nacelle direction SCADA signal (after applying the wake loss method). So the procedure is sequential rather than two different approaches being used. We have added a mention in the text that this correction introduces an uncertainty, where the largest contributor would be the reference direction obtained from the nacelle yaw position. As the yaw controller tracks the exact wind direction with a delay due to actuator limits, this approach is not ideal. But it was the only available method on the lead row where no hard targets were available. In future campaigns, we would use an aerial drone which is a new method we developed to address this problem.

10.     Line 99: Typographical error.

Corrected in text.

11.     Line 126: "The effect of the sea surface is included using a wall-stress model [...] At first, I found this to be confusing because as written it seems to imply that the effect of waves are considered (but that is just my interpretation), but this is a standard wall model. Consider rephrasing the sentence?

The author is correct that it is a standard wall model, wave effects are not included in the study. The sentence has been rephrased to avoid this confusion as follows

"The effect of surface friction is included using a standard wall-stress model, corresponding to a logarithmic velocity profile with a roughness length $z_0$ (Bou-Zeid et al., 2005) to represent marine boundary layers, without the need to resolve wave effects on the mesh."

12.      Sections 3.2 and 3.3: These sections (along with Appendices) are clear and detailed. Have the authors previously compared this AASM model to ALM, ADM, etc? There are a growing number of actuator turbine representations (e.g. actuator surface). It would be helpful to add a sentence or two to concisely summarize pros and cons of the selected AASM methodology.

While we have not compared the performance of the different models ourselves, we have now included a short discussion regarding the comparison between the different turbine models based on the reference we adapted our model from.

"The ADM is a simple representation of a wind turbine in the flow domain, in which a disc is used to parameterize the turbine forces on the flow. Though simple, the ADM does not accurately represent wind turbine operation due to a lack of discrete rotating turbine  blades, and hence often requires additional tuning to improve performance. Comparatively, the ALM provides a more accurate parametrization by representing the turbine blades as actuator lines, with each point exerting a force on the flow based on its 175 local inflow velocity and airfoil distribution. However, the ALM suffers from a limitation that the movement of the actuator line tip over a time step is limited to the size of a cell, thus requiring very fine time steps and hence high computational costs. To overcome this limitation, an Actuator Sector Model (ASM) was developed, which swept the rotor forces across a sector area and hence allowed for coarser time steps. The ASM therefore represents an intermediary between the ALM and ADM, with the ability to resolve structures in the near wake with greater detail than the ADM due to the presence of rotating  actuator lines,180 while avoiding the high computational cost associated with the fine time steps of the ALM (Storey et al., 2015)."

13.      Figure 4: I think this figure is really great, very clear. Why was a first-order time-filter selected? Has this decision been validated in previous papers/theses by the authors for the AASM? If so, please provide references and justification. Also, consider defining variables in the figure caption.

A first order time-filter was chosen for simplicity. We have not yet explored higher order time filters for the AASM, however this could be a topic for future work and has been remarked upon in the text.

14.      Figure 5: Are the deviations in this figure the result of a difference in the induction given the coarse grid resolution in SP-Wind?

Yes, the current grid resolution is restricted by the precursor database, which was originally developed for a bigger 10 MW turbine and a larger wind farm.

"Slight differences can be seen in both the simulated power and thrust, which can be attributed to the coarse grid resolution across the turbine blades in SP-Wind, which was was restricted due to the resolution of the precursor simulations used as they were originally developed for a bigger 10MW turbine and a larger wind farm. Using the same grid resolution for the smaller 2.3 MW turbines at Lillgrund leads to differences in induction at the rotor plane."

15.      Figures 6 and 7: I presume that the individual data points shown are not the full resolution of the precursor LES cases (e.g. CNk8 has two points in the boundary layer). Please clarify.

There are 225 data points in the vertical profiles, and the markers are plotted every 10 data points to avoid congestion in the figure. This is now stated in the captions.

16.      Section 4.1:

a. Does u^t represent the velocity vector (it is not bolded and no vector identification is included) or wind speed? If it is wind speed, is the spanwise velocity (v) neglected from this analysis? If it is a vector, are there unique values of u*^t z0^t for each of the streamwise and spanwise velocities?

u^t represents the velocity vector comprising of the stream-wise and spanwise velocities. The same friction velocity and surface roughness is used for both the velocity components. This has now been clarified in the text.

b. How do the authors replicate the wind veer from the LiDAR? This is not included in Eqs. (5) or (6).

We do not attempt to explicitly recreate the wind veer, and the veer is limited by the precursor database. As per their definitions, the PDBL has zero veer and the CNBL has limited veer at the rotor, as shown in Figure 7.

c. Also, e1 is not defined in Eq. (5).

Vector e1 represents a unit vector in the x direction, symbolizing that the off-set to the mean flow through a new surface roughness is applied only to the u velocity. This has now been clarified in the text.

d. This discussion and the references focus on surface layer similarity. Cases with geostrophic forcing at a finite Rossby number are also considered. Matching the friction velocity can lead to large errors in the geostrophic wind speed (as seen in Figure 8).

We have now included a discussion at the end of section 4.1 addressing the changes required when scaling the CNBL precursors. Essentially, scaling the CNBL with new friction velocities would imply changing the latitude and capping inversion strength of the boundary layer for the boundary layer height to remain constant.

17. Line 225: What does it mean for the range gate to span the hub height, as the hub height is one location.

This was an error and has been corrected to state range gates spanning 20% of the rotor area around hub height.

18. Table 3: What is the uncertainty associated with the TI estimation for both LiDAR and LES?

We have now included uncertainty estimates determined by the block bootstrap approach in Table 3. They are also presented here for completeness.

| Case | LiDAR TI [%] | LES TI [%] |
|------|------|------|
| $PDk_1$ | $5.52\pm0.09$ | $6.31\pm0.17$ |
| $PDk_2$ | $5.99\pm0.09$ | $6.27\pm0.14$ |
| $PDk_3$ | $6.64\pm0.11$ | $6.83\pm0.12$ |
| $CNk4_1$ | $5.12\pm0.17$ | $5.59\pm0.15$ |
| $CNk8_1$ | $5.86\pm0.16$ | $5.73\pm0.21$ |

19.      Figure 8 is messy with the inset figures overlapping the outer figures. Please consider reformatting. Also, how were the 95% CIs estimated for both LiDAR and LES? Please add this description to the text body.

Thank you for this suggestion, we have now split the figure in two to improve clarity. The block bootstrap method is used for evaluating the CIs, this information has been included in the caption.

20.      Section 4.2: Overall, I find this section and the outlined approach interesting. I recognize the technical challenges of what the authors are attempting, given fundamental inconsistencies between the CN or PD LES cases and the true ABL LiDAR measurements. Primarily, fitting neutral or PD ABL profiles to non-neutral ABLs may bias the results. So, I suggest the authors add an additional paragraph, addressing the following:
        a.   Discussion of the stability during the LiDAR measurement time periods.
        b.   Discussion of how the optimization process which finds the minimum error compared to conventionally neutral or pressure driven boundary layers (with no stratification) may bias the considered cases.
        c.   Given the poor agreement in the wind veer, discuss how the optimization process may have biased the results to over-emphasize wind speed compared to wind direction.
        d.   Discussion of how the pronounced disagreement between LiDAR and LES above the rotor area may bias the vertical transport into the wind turbine array atmospheric boundary layer (WTABL).
        e.   Finally, I find it a bit strange that several of the selected cases had LiDAR or loads measurements missing. Why did the authors not select other cases where all of the data were present (even if the precursor data differed slightly more than in other cases)?

We thank the reviewer for the suggestion and have now included an additional section 4.3 to acknowledge these shortcomings and sources of error. We agree that stability conditions at the Lillgrund site would have a significant effect on wind farm performance and the resulting errors in LES. However due to a lack of stability information during the measurement campaign, we were unable to include this in our methodology for recreating the inflow conditions. This has been summarized as below in section 4.3

"The developed optimization framework is therefore able to successfully recreate the inflow conditions at the Lillgrund wind farm for the chosen time periods, without significant errors in the mean inflow velocity across the rotor area, as evident by Figure 9. However, a few shortcomings of the optimization framework must be addressed. Atmospheric stability and stratification can play a significant role in the performance of a wind farm by affecting wake recovery and ambient turbulent intensity (Magnusson and Smedman, 1994). Additionally, due to the close land proximity of the Lillgrund wind farm, the atmospheric conditions at the farm can be significantly affected by a land–sea system as a combination of marine and land boundary layers. The current methodology does not take the meso-scale effects at the Lillgrund site into account, instead relying on a database comprising of marine pressure driven and neutral boundary layers to match the inflow conditions. This is due to the lack of atmospheric stability and temperature measurements at the Lillgrund site during the measurement campaign. If the data was available, the inflow database used could have been modified before the transformation procedure to consist of precursor boundary layers which best represent the stability conditions at the wind farm site. In its current form, the considered cases may be biased as they are chosen simply based on minimum error in the distance metrics and do not account for stability or land–sea effects. Additionally, without the inclusion of appropriate weights or additional distance metrics to penalize wind veer error, the current procedure may have been biased to overemphasize mean speed error as compared to wind veer error. The error in the LiDAR and LES velocity profiles above the rotor area could also affect the vertical transport into the wind turbine array atmospheric boundary layer (WTABL), which may affect the overall wind farm performance (Calaf et al., 2010)."

Unfortunately, the answer to point e is due to the timeline of the measurement campaign and the order in which data was made available. The LES simulations were started as soon as the inflow LIDAR and SCADA

data was available. However, the turbine loads and wake lidar data was made available at a much later date, when it was discovered that the selected time periods had significant missing measurements. Due to the high computational cost and shortage of time, we were unable to perform additional simulations considering different time periods. In future campaigns, care will be taken to make sure that all the data is available beforehand.

21.     Line 286: "This is in accordance with the errors in the inflow profiles observed [...]" The analysis of the power results in this sentence appears weak and selective. First, putting aside the power and looking just at the wind profiles in Figure 8, all of the PD cases have high error in the wind veer, not just PDk3. Second, it is not clear from Figure 8 that the PD cases have lower wind speed error than the CN cases. For example, CNk8 seems to have fairly low wind speed and wind veer error. Cases PDk1 and PDk2 do not appear to have significantly lower wind speed error compared to the LiDAR than CNk8. The authors should revisit this analysis with more attention. I suggest quantitative metrics to support the statements.

We thank the author for this remark, and revisited the error analysis with more attention. Upon further investigation, we discovered that the main influence on the power errors observed are due to the wind direction and the waked/un-waked state of the wind turbine. From the below figure, it can be seen that the turbines operating right behind the first turbines exhibit the highest power errors, while turbines operating in un-waked or towards the back of the farm exhibit lower errors. This indicates a difference in wake recovery in LES and the field measurements which is further confirmed in the wake recover analysis. The errors could be due to a difference in stability, the high errors above the rotor in the developed inflow boundary layers, or differences in meso-scale effects which can cause wind direction changes across the farm as is evident from the newly added figure 16. The higher power errors for the PDk3 case can also be attributed to a difference in controller at the low wind speed regime, where the PDk3 turbines within the farm are exhibiting high pitch angles and operating in a start-up regime with negligible power production, which is not happening for our turbines in the simulation environment. We have now included the following figure and a new paragraph in section 5.3 to discuss this.

[Figure]

22.     Figure 10: How sensitive are the bootstrapped CIs to the block length selected in the bootstrapping approach?

In a preliminary sensitivity study, it was found that the CIs did not vary significantly beyond block lengths of 600 seconds. The same was found to be true for number of bootstrap iterations beyond 1000.

23.     Figure 11 is challenging to see. Please modify the aspect ratio to be wider and shorter.

Thank you for the observation, we have now adjusted the figure as suggested and improved its visibility.

24.     Line 304: Do the LES turbines align themselves with the wind direction measured locally at each turbine?

A local wind direction controller is currently not included in the simulation methodology, and may affect the comparison against field measurements. However, local inflow angles at the turbines in LES were found to be negligible for the simulation cases, so the inclusion of a controller would not have had significant effects. Instead, we noticed a gradient in turbine orientation errors across the wind farm which indicates that there are meso-scale effects at the Lillgrund site which cause a changing wind direction across the farm, as can be seen in the following figure. We have now included this information in the text.

[Figure]

25.     Figure 13: I believe the y-axis of this plot should be 'Turbine orientation' or 'Nacelle position' rather than wind direction. In the field, the turbine's orientation is not always aligned with the wind direction, depending on the yaw controller dynamics. The local wind direction may also vary within the farm. Also, why are the turbine position errors high for all turbines in PD1? Does this indicate that the calculated mean wind direction was not correct?

We thank the reviewer for their suggestion, and have now updated the y-axis to 'Turbine orientation'. Upon further investigation, we discovered than an incorrect mean wind direction was used for the PD1 case when plotting figure 13. This has now been corrected in the text.

26.     Line 310: "This indicates [...]" This sentence seems to contradict the previous sentence.

This contradiction has now been removed and replaced with a more thorough investigation into the power errors.

27.     Figure 14: The difference between measurements and LES for the star symbols is not clear in these figures. The open star looks almost filled.

We thank the author for the observation, and have now increased the marker size of Figure 14 to improve visibility.

28.     Line 323: "For the PDk2 case [...]" Again, I encourage the authors to be more precise in their analysis. For PDk2, with the errorbars in Figure 14 representing 95% CIs, there are statistically significant errors between LES and LiDAR for B08 and B06, while the differences between LES and LIDAR for B07 are not statistically significant. This differs from the present analysis discussion in the paper.

The discussion in this section has now been split up to clearly talk about the flapwise moment errors and the fatigue load errors for clarity. While the flapwise moment errors for turbine B07 are not significant, for the fatigue loading they are significant.

29.     Figures 16 and 17: Why is the far wake comparison omitted from these plots? Please include. They are mentioned on line 335, so presumably the data is there. If they were selectively omitted due to poor agreement between LES and LiDAR, that is even more reason to include them in the figures.

A far wake analysis has now been included for the CNk4_1 case. This was not possible for the PDK_1 case as far wake lidar data was not available. The wake deficit figures have also been updated to include confidence intervals for the LES data.

[Figure]

**Figure 20.** Comparison of far wake velocity wake deficit at downstream locations from turbine C06 for the case $CNk4_1$. Line at y/D=0 is perpendicular to the turbine orientation from SCADA data. Shaded area represents 95% confidence interval on the mean.

30.     Line 337: "Observing the wakes [...]" I was not able to discern the turbine orientation story from the LiDAR results in figure 17(a). Could the authors rephrase this discussion to be more clear? Looking at figures 17(a) and 17(b), how is it clear that in LES there is partial wake while in LiDAR there is not, since only C05 can be seen (and it does appear to be in a partial wake).

This discussion has been rephrased to be more clear. In the PDK_3 wake deficit figure ,which has been now updated to include 95% confidence intervals on the LES data as well, it can be seen that the LES wake shifts more to the left than the LiDAR wake. As a result, the downstream turbine is more waked in the LiDAR case than in the LES case, leading to higher velocity and hence the higher observed power prediction.

---

## Author Comment (AC3)

**Response to reviewer 1**

This paper is an excellent piece of research, full of innovative ideas and techniques and very well written. It should definitely be published in WES. My suggestions below are not even minor revisions, but addressing them will clarify a few issues and hopefully improve readability.

I have one somewhat sad comment. To be honest, I was expecting a better performance of the entire modeling system and perhaps more wake analyses (only two turbine wakes were analyzed, qualitatively). I guess I should not be surprised because the data are never what we hope them to be. There were so many mismatches between the real wind farm and the simulated one, from pitch angles to yaw misalignment etc. But unfortunately, because of such data issues, I am not convinced of the goodness of either the SP-Wind solver with the AASM or the initialization technique using TotalControl Flow data. This is really sad because both are innovative and appear to be robust. Maybe a more in-depth data cleaning procedure could be applied to the dataset?

We thank the reviewer for the kind words and appreciation of the manuscript. The main goal of the current work was to highlight the challenges associated with detailed aeroelastic simulations of wind farms and comparison against a rich field data set which includes SCADA, LiDAR and Loads comparisons. Through the detailed comparison, we were able to highlight the areas of numerical modelling which still require improvement for accurate representation of wind farm performance. While some data cleaning was conducted to remove channels with high noise or missing data, it was not clearly evident that the remaining errors are due to errors in measurements or a real feature due to the differences between our modelling approach and the real conditions at the wind farm.

Please find below responses to the reviewers point comments, which we believe have improved the quality and clarity of the manuscript.

1.      Either "dataset" or "data set" should be used, but not "data-set". Please replace it throughout.

This suggestion has be incorporated in the manuscript and data set is now used consistently throughout.

2.      Line 20: you should expand your literature review because citing only two studies about LES applications to wind farms is just not acceptable. There are so many more such studies, including several about Lillgrund. Here is an incomplete list:

Archer et al. (2013), Bhaganagar and Debnath (2015), Calaf et al. (2010, 2011), Chaudhari et al. (2017), Churchfield et al. (2012), Fleming et al. (2014), Ghaisas and Archer (2016), Ghaisas et al. (2017), Han et al. (2016), Lu and Porte-Agel (2011), Martínez-Tossas et al. (2015), Meyers and Meneveau (2011), Xie and Archer (2017)

We thank the author for a comprehensive list of references, and have now included additional references in the text for LES and aerodynamics of wind farms.

3.      Lines 22-23: you should not list a series of issues that LES has apparently helped address without citing the literature studies that you think did it. You need to add at least one citation for each of these terms: gusts, atmospheric stratification, and turbine-wake interactions. I assume that Mehta at al. (2014) was about local wind climate only.

The paper by Mehta et al (2014) provides a literature review of aerodynamics of wind farm LES, which further expands upon the mentioned effects. As the issues are not explored in our work and for the sake of brevity, we chose to include a reference for the review for completeness. We have now also included an additional review paper (Porte-agel 2020) for wind farm flows.

4.      Around line 190: What exactly is a "Pressure-driven" boundary layer? Is it neutral? It seems that there is no thermal inversion in those. What is a "conventionally neutral" BL? Please add some more description about the two datasets used (PDBL and CNBL).

A PDBL is a neutral boundary layer which ignores the Coriolis effects and free atmosphere stratification, whereas a CNBL includes these effects and is characterized by a neutral boundary layer capped with stable free atmosphere. Both the PDBL and CNBL are neutral in the surface layer. This has been clarified in the text in section 4.1 in the following lines, and a reference has been included which details the generation of the precursor datasets.

" The TotalControl flow fields are initialized using mean velocity profiles upon which random divergence-free perturbations are added. These initial conditions are then advanced in time for 20 physical hours, so that the influence of the unphysical perturbations has disappeared, and the flow has reached a fully turbulent and statistically stationary state. The PDBL is a simple representation of a neutral atmospheric boundary layer which ignores the effects of rotation and thermal stratification, while CNBL provides a more realistic representation by including these effects. Both these boundary layer types have been extensively used in wind-farm LES."

5.      Related to question 2 above, it would really help is the same information could be provided for all 5 cases (PDBL and CNBL). In Figure 6, it would be great if you could add the same profiles but for CNBL. In Figure 7, it would be even better if you could add the profiles for the PDBL, especially the potential temperature profiles.

Both the boundary layers in the dataset are fundamentally different, hence it was not possible to have similar figures for all the 5 cases. This is reason why the 2 PDBL were grouped together in Figure 6, and the 3 CNBL in Figure 7. The PDBL are unidirectional and do not include temperature effects, therefore it is not possible to plot veer and potential temperature effects as was done for the CNBL profiles.

6.      Figure 6a: is the value of z0 reported for PDkhi correct? It is inconsistent with the value in Table 1.

This was an inconsistency and the value in Table 1 has now been corrected.

7.      Lines 200-205: It is brilliant to re-scale the precursor runs from Total Control Flow via appropriate z0 and u*. In the text, it sounds like u* is "imposed" in the Total Control Flow fields (together with z0). Is that true? Can u* be imposed during initialization? I am just curious. If so, please add the values of imposed u* in Table 1 in another column.

The following lines have been added in section 4.1 to clarify these points

" All the boundary layers have been intialized using a friction velocity of $u_*$=0.28 m/s , which is a typical value for marine boundary layers. For the PDBL, this requires a driving pressure gradient of $\nabla p_\infty / \rho =$ -5.2267×10$^{-5}$ m/s$^2$ for a boundary layer height of 1500 m according to the equation, $u_* = \left( -\frac{H}{\rho} * \nabla p_\infty \right)^{\frac{1}{2}}$ "

The pressure gradient in the Lillgrund simulations is adapted for the new values of friction velocities required for the transformation. As all the flow fields have the same friction velocity, we include this information as a line in the text instead of another column in the table.

8.      (5): what is vector e1?

Vector e1 represents a unit vector in the x direction, symbolizing that the off-set to the mean flow through a new surface roughness is applied only to the u velocity. This has now been clarified.

9.      Table 1: Is z0 for case PDkhi really 2 x 10^-5 m? In Figure 6a, it appears to be 2 x 10^-3 m. There is an inconsistency.

This was an inconsistency. z0 for the case PDki is 2 x $10^{-3}$ m, and has now been fixed in the table.

10.     Page 13: This idea is really cool, but the notation is tough. For example, u and v are usually the two horizontal components of the wind vector. Here, they are used to indicate wind speed, but u for LES and v for Lidar. A bit confusing. X and Y are both wind speeds, but usually x and y are Cartesian coordinates. The overbar indicates what exactly? A 75-minute time average at each point? Or is there some sort of horizontal average first to obtain "time average profiles at range gate locations", which are then weighted with w? Why is it so important to minimize the covariance distance? Given the relatively large errors in wind direction (Figure 8b, especially PDk3), I was thinking that maybe a wind direction distance could be minimized instead? And finally why is the simple sum of the two distances chosen? Isn't wind speed more important? These are all requests to add some clarifications in the text.

We thank the reviewer for the suggestions and have reworded this section to improve clarity. X and Y are now instead replaced with P and Q to avoid confusion. The overbar indicates a 75 minute time average. Before averaging, data is extracted from the LES domain at points equivalent to the LiDAR measurement range gate locations. Along with the mean velocity profiles, the covariance distance is also minimized to aid in the selection of transformation parameters which gave similar turbulence intensity as the LiDAR measurements. Without including the covariance distance, we observed that while the mean profile could match well, the turbulence intensity can have large differences. Figure 8b does not show differences in wind direction, but in wind veer across the rotor area. All the PDBL simulations have no spanwise velocities, hence zero veer. This emphasizes that the PDBL is not a realistic representation of the atmospheric boundary layer, as can be seen in figure 8b where the field measurements have significant veer across the rotor area. Since the optimization framework minimizes the distances between both the u and v velocity components, the wind direction is implicitly minimized as well and does not need to be including separately. We agree that choosing a direct sum might be too simplistic, and in future work a parametric study could be conducted to determine which metric has more significance when determining similarity. These points have now been clarified in the text in more detail.

11.     Table 3 is not cited anywhere in the manuscript. Perhaps at line 253?

A reference to Table 3 is now included in section 4.2

12.     Line 2054: This domain is huge! Why was such a large domain chosen? I thought that perhaps that was the domain size in the PDBL and CNBL datasets (if so, please mention it). If not, then how do you "extract" a portion of the data from those datasets to match your domain?

The domain size in the current simulations is indeed restricted by the domain size of the precursor datasets, which was originally designed for a larger wind farm and this has now been clarified in the text in the following lines in section 5.1

"The choice of domain size is restricted by the one used for the precursor data sets, which was initially designed for simulations with the much larger TotalControl reference wind farm to avoid blockage effects"

13.     Line 277: this phrase does not make sense … What does "made to advance" mean? Who advances what? What do you mean by "the inflows"? I thought you took initial and boundary conditions from TotalControl Flow, after correcting them with the scaling parameters. Please rewrite to clarify.

These lines aim to explain how the concurrent precursor methodology of windfarm LES has been modified to include flow transformation for representing the inflow conditions at the Lillgrund wind farm. The entire three-dimensional instantaneous flow field from the TotalControl precursor database is used as the initial starting conditions for a precursor simulation, which are then advanced in time using the Runge Kutta

scheme. A slab is taken from the precursor domain (without turbines), transformed by the scaling parameters to match the inflow conditions at the Lillgrund wind farm, and added to a second domain which contains wind turbines through a fringe region. This has been now clarified in the text in section 5.1.

"First, the inflows from the previously generated TotalControl precursor database are made to advance in time in a domain without wind turbines, called the precursor domain. Concurrently, the flow is transformed using the identified transformation parameters through the optimization framework to obtain the five cases identified in Table 2 and fed into a second domain which contains wind turbines as body forces $F_R$ through a fringe region"

14.     Line 301: It sounds like you should have cleaned up the SCADA data and removed high-pitch cases. Why did you not do it?

It was not directly clear whether the high pitch cases were a result of corrupted data, or a feature of the controller operational on the Lillgrund turbines. Hence it was not straightforward to remove these cases. As the pitch values were not unrealistically high, it could be that the Lillgrund turbines exhibit some degree of pitching in region 2 as well, which was not replicated in our controller implementation. In hindsight, it would've been better to screen the data to learn about such issues in advance and this can be incorporated into future workflow.

15.     Figures 16 and 17: I assume that the orange lines correspond to the location of the vertical profiles. Did you also average along those lines or did you pick the wind speed at the exact points downstream at 0.5D, 1D etc?

Figures 16 and 17 represent wake deficit in the horizontal plane behind the turbine at different downstream locations, and not the vertical profiles. The horizontal profiles were compared at a cross section spanning 2 rotor diameters at the same points behind the rotor in the LES results and the LiDAR wake measurements.

---

## Author Response (AR2)

**Author response – Minor revisions**

Overall, I am impressed by the ambition of this paper which combines many years of incremental progress in improving LES simulations of wind farms into a single comparison between LES and field data. My main takeaway from the predictive accuracy of the LES is that progress has been made, but there is a lot of room for improvement (e.g. meso-microscale coupling rather than simplified boundary layers which approximate the wind speed profiles), as there are significant differences between LES and field data in both power and loads for most turbines within the farm. I hope in their final revisions, the authors can help contextualize how their results can be useful to guide future research.

We thank the reviewer for taking the time to review the manuscript, and acknowledging the importance of the performed study. Through the detailed analysis performed in this work, we aim to highlight the strengths and weakness of LES solvers in predicting performance of offshore wind farms. We have taken the reviewer's valuable feedback into consideration, and have made the following changes as per the point comments.

Point comments:

I appreciate that the authors have added much more quantitative analysis and discussion in their revised paper, I think that adds clarity to the results. Yet, I believe the following statement in the abstract is still not an accurate summary of the results presented in this paper.

"Nevertheless, statistically significant errors were observed on an individual level for a few turbines deeper in the wind farm operating in wakes across the simulated cases."

As per the reviewer's suggestion below, we have performed an analysis to quantify how many turbines for each case have statistically significant errors. We observed that this proportion was higher for the cases which have a larger number of aligned turbines, and in general it was the waked turbines that exhibited statistically significant errors. The above line the abstract is thus modified to be "**However, when comparing individual turbine power production, statistical significant errors were observed for 16% to 84% of the turbines across the simulated cases, with larger errors being associated with wind directions resulting in configurations with aligned turbines**"

Taking a look a Figure 12 (I have counted by hand so I may be off by a couple), 32 of the total 48 turbines (32/48, or 67% of the total turbines) have statistically significant differences (error bars which don't overlap) between the LES power predictions and the field data. I encourage the authors to perform this quantitative analysis more formally than I just have, and to add these quantitative results to the abstract.

We thank the reviewer for this suggestion and have performed this quantitative analysis for all the cases. The results are summarized in the table below and have been included in the manuscript in section 5.3.

| Case | Turbines with significant errors [%] |
|------|--------------------------------------|
| $PDk_1$ | 58.3 |
| $PDk_2$ | 16.6 |
| $PDk_3$ | 83.3 |
| $CNk4_1$ | 25.0 |
| $CNk8_1$ | 35.4 |

I want to again emphasize that I believe this paper and the methods are of high quality and are a useful contribution to the literature. But I don't think we can overstate the fidelity of agreement between high quality microscale LES and field data (recommendations also echoed by the other reviewer). There appears to be a long way to go before we have reliable LES predictions of wind farm power (much research to be done!). This paper can be an excellent summary of key technical gaps.

We thank the reviewer for their recognition of the research and methods presented in the manuscript. We have included the following lines in the conclusion section to highlight the technical gaps highlighted in the paper. "**The analysis in this work thus highlights certain areas in wind farm LES which still require further research to make accurate performance predictions for offshore wind farms. In particular, lack of prior knowledge of meso-scale effects, field turbine control logic, and insufficiently fine grid resolution are highlighted as significant actors which can lead to errors in predictions. Therefore, future work should focus on improving on these effects , in particular for situations with aligned wind turbines**"

Section 4.3 is well-written and useful!

We thank the reviewer for their kind words.

Line 373: As in the first comment, this sentence is worth rephrasing from "good agreement with field data" to "reasonable qualitative agreement with field data, [...], but statistically significant differences for 32/48 of the turbines considered."

As per the performed analysis, this line has now been modified to "**However when comparing individual power production on a turbine level, statistically significant errors are evident for 58% of the turbines for the PDk1 where error bars for the 95% confidence intervals do not overlap when comparing LES predictions and field measurements. The same analysis is performed for all the simulated cases, and the proportion of turbines with statistically significant errors are presented in Table 5. In general, it was observed that the turbines at the front of the farm did not have statistically significant errors, however turbines operating within the wakes of an upstream turbine had higher errors which were significant for wind directions resulting in an aligned configuration**."